ecology, environmental science

pyrodiversity, biodiversity, functional diversity, wildfire, landscape ecology, fire ecology

**Author for correspondence:**
Zachary L. Steel
e-mail: zlsteel@berkeley.edu

# Quantifying pyrodiversity and its drivers

Zachary L. Steel[1], Brandon M. Collins[2,3], David B. Sapsis[4] and Scott L. Stephens[1]

[1]Department of Environmental Science, Policy and Management, University of California–Berkeley, Berkeley, CA 94720, USA
[2]Center for Fire Research and Outreach, University of California, Berkeley, CA 94720, USA
[3]USDA Forest Service, Pacific Southwest Research Station, Davis, CA 95618, USA
[4]California Department of Forestry and Fire Protection, Sacramento, CA 95814, USA

ⓘ ZLS, 0000-0002-1659-3141

Pyrodiversity or variation in spatio-temporal fire patterns is increasingly recognized as an important determinant of ecological pattern and process, yet no consensus surrounds how best to quantify the phenomenon and its drivers remain largely untested. We present a generalizable functional diversity approach for measuring pyrodiversity, which incorporates multiple fire regime traits and can be applied across scales. Further, we tested the socioecological drivers of pyrodiversity among forests of the western United States. Largely mediated by burn activity, pyrodiversity was positively associated with actual evapotranspiration, climate water deficit, wilderness designation, elevation and topographic roughness but negatively with human population density. These results indicate pyrodiversity is highest in productive areas with pronounced annual dry periods and minimal fire suppression. This work can facilitate future pyrodiversity studies including whether and how it begets biodiversity among taxa, regions and fire regimes.

## 1. Introduction

Fire is a fundamental ecological process [1] that plays a central role in biome distribution [2], ecosystem process [3] and biodiversity globally [4]. Fire patterns and their ecological consequences differ according to a number of important fire regime characteristics including burn frequency, severity, seasonality and spatial pattern [5]. Much effort has gone into quantifying the central tendencies of these fire regime characteristics (e.g. mean fire return interval) and their underlying drivers [6,7], but until recently what determines the variation of fire regime characteristics, known as pyrodiversity, has received little attention. Martin & Sapsis [8] first proposed that pyrodiversity begets biodiversity by creating heterogeneous landscapes composed of dissimilar habitats and ecological niches. Since the theory was formalized, the potential importance of heterogeneity in fire regimes for ecosystem pattern and process has gained increasing attention both in research and ecosystem management [4,9]. However, the expanded consideration has come with little consistency in the definition or application of the pyrodiversity concept. A generalizable approach for quantifying pyrodiversity and an improved characterization of the phenomena's socioecological drivers is necessary for advancing the understanding of its ecological importance.

The presumed link between pyrodiversity and biodiversity has influenced conservation efforts, particularly where prescribed burning or 'patch mosaic burning' is used to diversify fire histories across a managed landscape [9]. However, the development of robust ecological linkages to pyrodiversity has been hampered by our limited ability to fully capture relevant fire history components with sufficient spatial resolution and temporal extent. This limitation has been mitigated in recent years by advances in computing capabilities and spatial data availability [10]. For example, the 'visible mosaic' represented by the landscape pattern created by the most recent wildfire and subsequent successional processes can be easily observed [3,11]. However, observing the

'invisible mosaic' that includes components of fire history such as the timing and severity of previous fire events requires access to decades of remotely sensed fire histories. Assessing the relative importance of these legacy effects may be necessary for effective conservation and ecosystem management in fire-prone areas [9,12].

The complexity associated with distilling relevant fire regime components (subsequently referred to as 'traits') into a measure of pyrodiversity has resulted in varied approaches. Often these methods have focused a single fire regime trait such as burn severity [13,14] or frequency [12,15]. Such approaches implicitly assume a single trait serves as a surrogate for other fire regime characteristics and captures the most relevant aspects of pyrodiversity [4]. This is likely a valid assumption in some cases, but without an understanding of how fire regime traits covary this can result in misleading conclusions [16]. Other studies have incorporated multiple traits and treated unique combinations as distinct aggregates or 'species' when applying biodiversity metrics such as Simpson's diversity index [17]. However, traditional diversity metrics do not account for the trait distance between species and in the case of fire histories, definitions of species are sensitive to how continuous measures are classified into levels (e.g. four or more classes of burn severity). Hempson *et al.* [18] proposed perhaps the most general method of assessing multiple dimensions of pyrodiversity by calculating the convex hull (functional richness) of four fire-level traits, which do not capture within-fire variation described by burn severity and patch size. Together these assessments and others provide valuable contributions to our understanding of pyrodiversity's ecological role and their complementary strengths provide the scaffolding for a more comprehensive approach necessary to test how pyrodiversity influences biodiversity and other processes and whether these relationships vary among taxa and regions.

Fire regime central tendencies are controlled by climate, topography and human influence [6,19], and are reciprocally dependent on the structure and flammability of extant vegetation [2]. Through the annual and seasonal availability of solar energy and water balance, climate determines distributions of vegetation types, primary productivity and fuel flammability [7,20]. Topography also influences water balance, but can further exert direct control on fire behaviour [5], which in the aggregate likely influences fire patterns across landscapes [21]. Humans have influenced wildland fire for millennia either through direct management, accidental ignitions or indirectly through alterations of vegetation via land-use change [19,22]. In many areas, the nature of human influence has shifted from indigenous fire use that was locally driven and variable across landscapes to contemporary broad-scale fire management (dominated by suppression) that has homogenized landscapes [23,24]. These underlying drivers likely influence variation in fire patterns as well, either directly or as mediated by total burn activity.

Here, we build on Martin & Sapsis's [8] original definition of pyrodiversity and previous approaches of measuring the phenomenon (especially Hempson *et al.* [18] and Ponisio *et al.* [17]) to develop a general method for quantifying pyrodiversity using four fire regime traits within a functional diversity framework. We apply this measure of pyrodiversity broadly across all forested areas in the western United States and assess how pyrodiversity varies with climate, topography and human influence. This approach is fully reproducible, can be applied at local or landscape scales using associated code, and may help advance our understanding of the role of pyrodiversity in the maintenance of biodiversity and ecological function.

## 2. Material and methods

### (a) Generation of trait surfaces

We used four fire regime traits to calculate contemporary pyrodiversity: (i) fire return interval (frequency), (ii) burn severity, (iii) burn season and (iv) patch size (figure 1*b*). These traits are commonly used to define fire regime groups, are important determinants of ecosystem process in fire-adapted systems [5,6], and follow the original characteristics of pyrodiversity [8]. We mapped each of the four fire regime traits across the western United States using fire perimeter data from the national Monitoring Trends in Burn Severity database, which includes all large fires (greater than 404 ha) in the region between 1985 and 2018 [26]. Fire return intervals were calculated as the difference between burn years of overlapping fire perimeters, as well as the first and final year of the dataset. Thus, fire frequency in this case should be interpreted as a minimum estimate useful for relative comparisons rather than absolute assessments of frequency. Burn season was determined by the recorded ignition date and was transformed to the cosine of radians to account for the cyclical nature of the date (i.e. so that the last and first day of the year are consecutive). Burn severity was estimated in units of composite burn index (CBI) for each fire using Landsat imagery (TM and OLI sensors; 30 m resolution) and Google Earth Engine following Parks *et al.* [10]. CBI is a field-based measure of fire effects on vegetation, which is commonly estimated using remotely sensed imagery for landscape-scale assessments [27]. Values of CBI range from 0 to 3, which represent no change to complete mortality of above-ground vegetation. Currently, the modelled relationship between remotely sensed fire severity estimates and ecosystem impacts on the ground are more robust in forests than other cover types [10,28]. We calculated patch size by defining distinct patches in each burn year using the CBI categories of unchanged, low-, moderate- and high-severity as defined by Miller & Thode [27]. To allow for reasonable computation time, the precision of frequency, seasonality, severity and patch size were limited to years, tenth cosine radians, 0.5 CBI and log hectares, respectively.

When calculating contemporary fire regime traits, values are often averaged across a period of record or only the most recent fire event is used. For example, fire frequency could be quantified as the mean of inter-fire intervals since reliable records began or the time since the previous fire [29]. Both options are sub-optimal if the phenomenon of interest is most sensitive to recent events but previous fires (the 'invisible mosaic') maintain some influence over landscape pattern and process [12]. We bridge these extremes by implementing a flexible recency-weighted averaging approach when calculating pixel-wise trait values. Trait values from recent fires (or intervals) receive the greatest weight with the weight or importance of earlier events decaying with order. Here, we assigned a decay rate of 0.5, for which each prior value receives half the weight of the more recent. A decay rate of 0 can be specified if all events are assumed equally important or 1 if only the most recent event is considered. We chose to weight by fire order rather than time to avoid confounding weighting and the fire frequency trait being measured.

### (b) Pyrodiversity calculation

We calculate pyrodiversity using a measure of functional dispersion (FDis) defined by Laliberté & Legendre [25]. FDis is analogous to the univariate weighted mean absolute deviation. It is independent of species richness [25], which is preferable when the boundaries between species are unclear and the number of

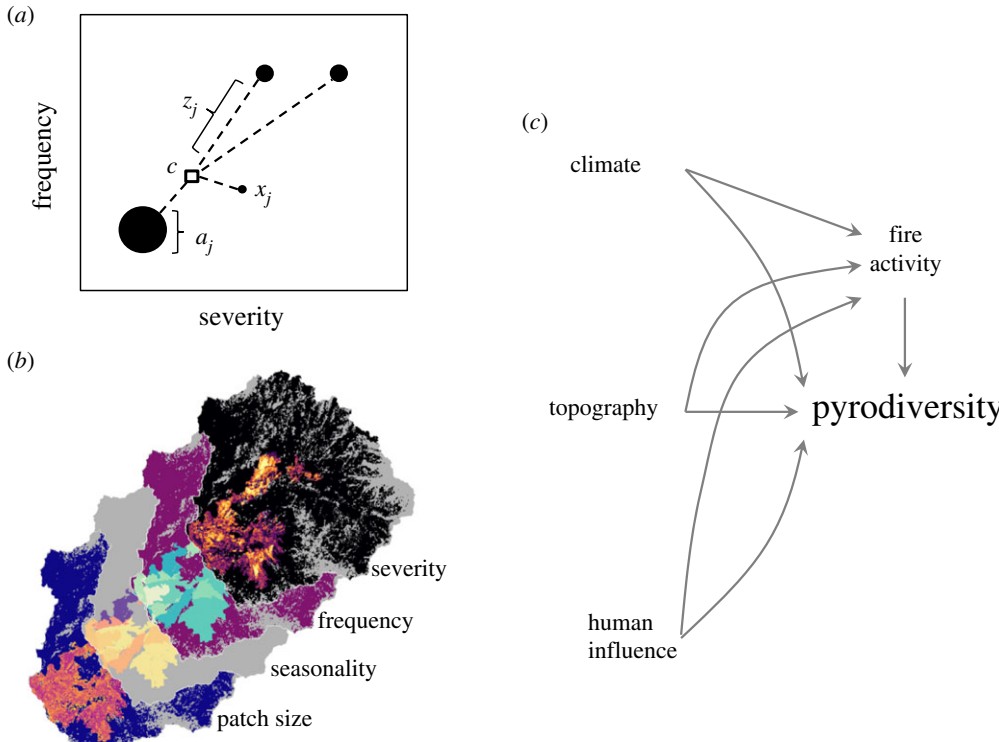

**Figure 1.** (*a*) A simplified example of how pyrodiversity is calculated using the functional dispersion metric (FDis), adapted from Laliberté & Lengendre [25]. $x$ represents the location of $j$ unique fire histories (species) in multi-dimensional trait-space, $c$ is the trait-space centroid of a landscape (community), $z_j$ is the trait distance of history $j$ from $c$ and $a_j$ is the frequency (abundance) of history $j$. FDis is calculated as the weighted mean distance from $c$. (*b*) Fire trait surfaces used to calculate pyrodiversity for an example watershed. (*c*) Conceptual model of the ultimate and proximate drivers of pyrodiversity. (Online version in colour.)

species varies among communities. FDis measures the mean multi-dimensional distance of unique species from the centroid of a community, weighted by abundance (figure 1*a*). FDis is unitless, bounded by zero on the low end, and is theoretically unbounded on the high end. In the case of pyrodiversity, each grid cell (30 m resolution) is considered an individual, unique combinations of fire regime traits (fire histories) are considered species, abundance is the frequency (number of pixels) of each unique history and a landscape is considered the community of interest. A functional diversity approach is an improvement on more traditional measures of diversity (e.g. richness and Simpson's diversity) because it incorporates information about the distance of individuals in multi-dimensional trait-space rather than assuming each unique combination of fire histories are equally and fully distinct. For example, when using Simpson's diversity index, two cells burned by the same fires but with slightly different severity would be considered unique, as well as equally dissimilar from a pixel with no recent fire history. Functional richness, as measured by the volume of the minimum convex hull, can also be a useful metric of pyrodiversity [18], but is sensitive to outliers and variable sample sizes (electronic supplementary material, [25]). Functional trait metrics such as functional richness and FDis allow for differential weighting of traits [25], which facilitates explicit testing of the relative importance of different components of pyrodiversity and mechanistic relationships [30]. Here, we weight the four pyrodiversity traits equally and rely on future applications of this method to test and parameterize the importance decay rate and relative trait weights for the ecosystem and processes of interest. Trait rasters created with the 0.5 decay rate can be found at https://figshare.com/articles/Pyrodiversity_westCONUS/12478832 and code is available at https://github.com/zacksteel/pyrodiversity for generating custom trait surfaces for future research. These data can be used to calculate pyrodiversity either across broad extents as demonstrated here or locally (e.g. around field survey locations).

## (c) Pyrodiversity trait covariance

While FDis accounts for redundancy among traits through ordination [25], understanding how fire traits covary is valuable for categorizing fire regime groups, as well as assessing the mechanisms by which variation in fire traits affects ecosystem pattern and process. We calculated correlations among the four pyrodiversity traits at the watershed scale. To test whether correlations varied with the amount of recorded fire history, we progressively filtered out less frequently burned watersheds with increasingly higher thresholds of number of fires recorded. Specifically, correlations were made among traits for all watersheds with minimum number of fires ranging from 1 to 15.

## (d) Pyrodiversity drivers

We assessed the hypothesized ultimate drivers of climate, topography and human influence on pyrodiversity using a (i) pyrodiversity model and a (ii) burn activity model. These models represent direct and indirect (burn activity-mediated) effects, respectively (figure 1*c*). We model direct effects on pyrodiversity as

$$\text{pyrodiversity}_{i,j} \sim \text{Beta}(\bar{P}_{i,j}, \theta)$$

$$
\begin{aligned}
\text{logit}(\bar{P}_{i,j}) = {} & \alpha_0 + \alpha_j \\
& + \beta_{\text{AET}} * X_{1,i} + \beta_{\text{CWD}} * X_{2,i} + \beta_{\text{AET}*\text{CWD}} * X_{1,i} * X_{2,i} \\
& + \beta_{\text{elev}} * X_{3,i} + \beta_{\text{rough}} * X_{4,i} + \beta_{\text{elev}*\text{rough}} * X_{3,i} * X_{4,i} \\
& + \beta_{\text{pop.den}} * X_{5,i} + \beta_{\text{wild}} * X_{6,i} \\
& + \beta_{\text{prop.burn}} * X_{7,i} + \beta_{\text{prop.burn}^2} * X_{8,i}
\end{aligned}
$$

$$\alpha_j \sim \text{Normal}(0, \sigma_{\text{HUC2}})$$

$$(2.1)$$

where actual evapotranspiration ($\beta_{\text{AET}}$), accumulated climate water deficit ($\beta_{\text{CWD}}$) and their interaction ($\beta_{\text{AET}*\text{CWD}}$) are estimates of

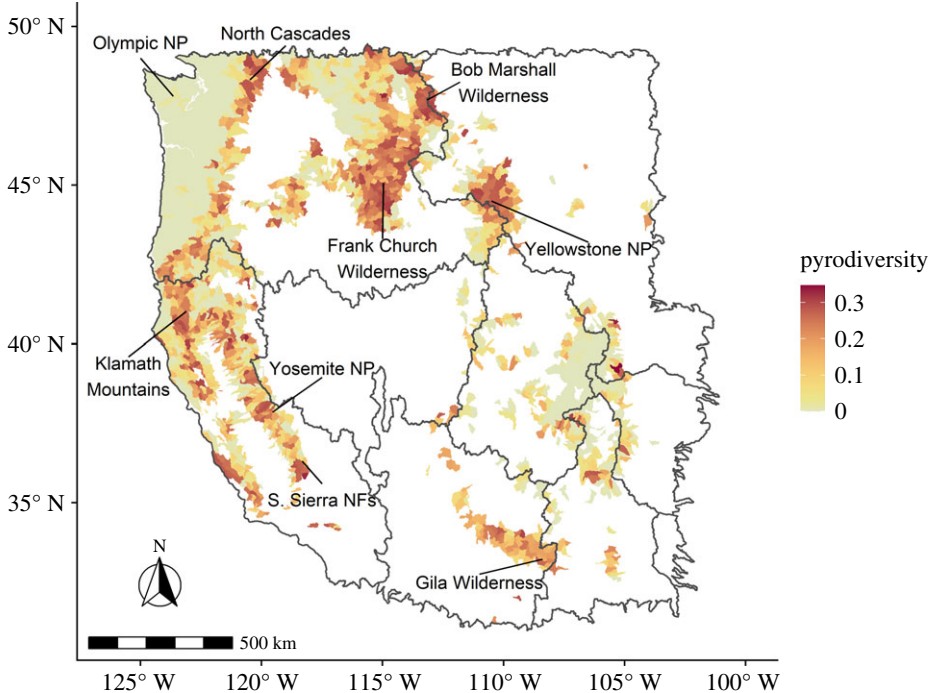

**Figure 2.** Pyrodiversity of forested watersheds (HUC10s) in the western United States. Watersheds with less than 50% forest cover were not evaluated and are shown in white. The broader-scale HUC2 watersheds (clipped to the region of interest) are shown as black outlines. (Online version in colour.)

climatic effects. Elevation ($\beta_{\mathrm{elev}}$), roughness ($\beta_{\mathrm{rough}}$) and their interaction ($\beta_{\mathrm{elev*rough}}$) are topographic effects. Population density ($\beta_{\mathrm{pop.den}}$) and proportion of watershed $i$'s land area in wilderness ($\beta_{\mathrm{wild}}$) are surrogates for human influence. We hypothesize much of the effects of these ultimate drivers are mediated by burn activity, here represented by the proportion of flammable area burned between 1985 and 2018 ($\beta_{\mathrm{prop.burn}}$) and its quadratic ($\beta_{\mathrm{prop.burn^2}}$). This metric is cumulative and can exceed one in the case of multiple burns in the same area. Our sample unit $i$ are forested watersheds delineated by the 10-digit Hydrologic Unit Code (HUC10; median area = 755 km$^2$). We account for spatial structuring of these units by including larger watershed $j$ (two-digit HUC2s; median area = 437 000 km$^2$), within which HUC10s are nested, as varying intercepts $\alpha_j$. In total, we assessed 1971 watersheds and 3306 fires.

To quantify indirect effects on pyrodiversity, we modelled proportion burned area as a function of the same climate, topographic and human influence variables as in equation (2.1), excluding $\beta_{\mathrm{prop.burn}}$ and $\beta_{\mathrm{prop.burn^2}}$ (electronic supplementary material, equation S1; figure 1c). This burn activity model is linked with the pyrodiversity model via a Bayesian multi-variate and multi-level model using the brms and rstan packages in R [31–33]. The multi-variate model allows us to predict the direct and indirect effects of the ultimate drivers and quantify their combined effect while properly propagating uncertainty through the model chain. In this way marginal effects are estimated by first fitting the burn activity model to generate a posterior distribution of proportion burned area and subsequently incorporating this full distribution as predictors of $\beta_{\mathrm{prop.burn}}$ and $\beta_{\mathrm{prop.burn^2}}$ in the pyrodiversity model. All predictor variables are standardized with a mean of zero and standard deviation of one. Model code, data and additional methodological details can be found in the electronic supplementary material.

## 3. Results

Watersheds experienced a wide range of fire activity during the study period, with a median of two fires (mean = 3; range: 0–48). These fires resulted in a median of 3.6%

(mean = 16%; range = 0–250%) of the flammable area burned. The median watershed had a pyrodiversity value of 0.04 (mean = 0.09; range = 0–0.35). Hotspots of pyrodiversity include watersheds in the North Cascades, the Northern Rocky Mountains within and around the Frank Church–River of No Return Wilderness, Yellowstone National Park, the Mogollon Rim including the Gila Wilderness, and the mountainous regions of California especially the Klamath Mountains, and parts of the Sierra Nevada (figure 2).

When including watersheds with little recent fire history, variation in burn frequency, patch size and severity are highly correlated. However, when sequentially excluding areas with less active fire histories, these correlations quickly dissipate. The correlation between frequency and patch size approximates 0.5 when considering watersheds with 14 or more fires since 1985. The frequency–severity correlation drops below 0.5 once watersheds with fewer than eight fires are excluded. Patch size and severity plateau at approximately 0.65 when considering watersheds with 10 or more fires. Seasonality is largely uncorrelated with the other three fire regime traits, starting between 0.13 and 0.23 when watersheds with at least one fire are included, and dropping below or near zero when restricting correlations to areas with more active fire histories (figure 3).

Climate, topography and human influence metrics show clear effects on proportion of flammable area burned 1985–2018. Proportion wilderness followed by climatic variables showed the strongest effects. Proportion burned area increased with proportion wilderness with a scaled effect ($\beta_{\mathrm{wild}}$) of 0.70 (90% confidence interval [CI] = 0.53, 0.86). Climatic water deficit (CWD) also had a strong positive effect ($\beta_{\mathrm{CWD}}$ = 0.53; CI = 0.47, 0.59), as did actual evapotranspiration (AET; $\beta_{\mathrm{AET}}$ = 0.10; CI = 0.048, 0.16), and the interaction of CWD and AET ($\beta_{\mathrm{AET*CWD}}$ = 0.27; CI = 0.23, 0.31). Both topographic roughness ($\beta_{\mathrm{rough}}$ = 0.15; CI = 0.11, 0.19) and elevation ($\beta_{\mathrm{elev}}$ = 0.084; CI = 0.011, 0.16) are positively associated with burn area, but these variables interact negatively

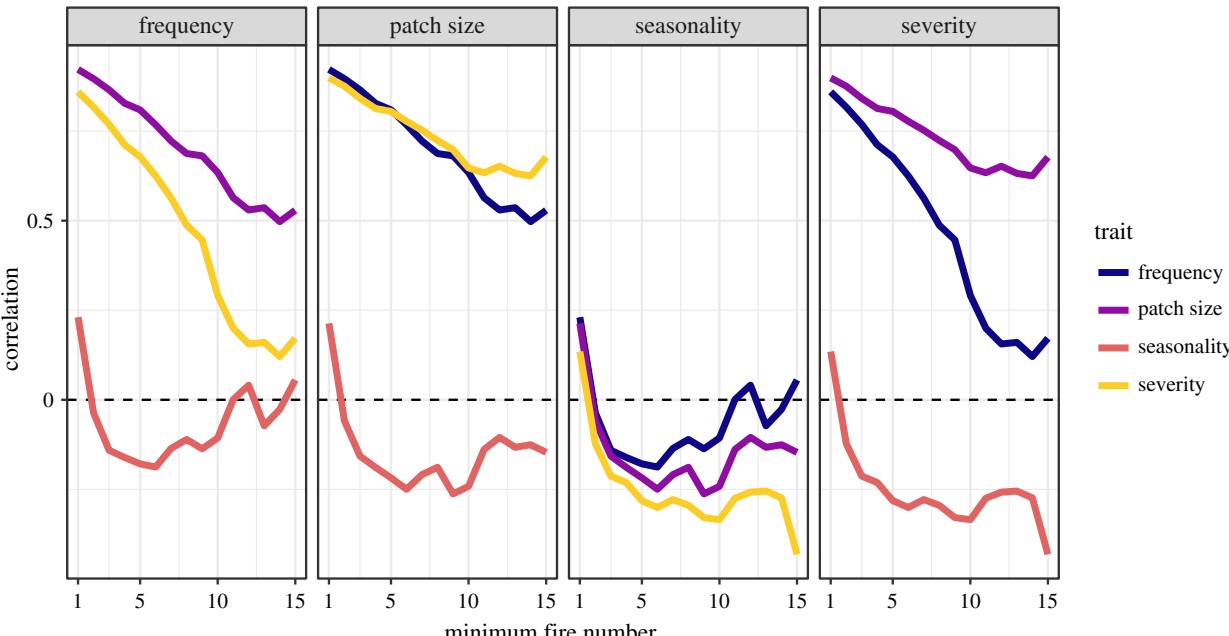

**Figure 3.** Correlations among watershed-level fire regime trait dispersion. Comparisons were made across a range of minimum fire numbers by sequentially removing watersheds with fewer recorded burns between 1985 and 2018. (Online version in colour.)

($\beta_{\text{elev}*\text{rough}} = -0.10$; CI = $-0.14$, $-0.068$). Human population density was negatively associated with proportion burned area with an effect estimate ($\beta_{\text{pop.den}}$) of $-0.15$ (CI = $-0.183$, $-0.109$) (electronic supplementary material, table S1).

When the proportion of flammable area burned is included as a predictor of pyrodiversity it has by far the greatest effect, with much of the ultimate effects of climate, topography and human influence being mediated by this variable. The proportion burned area is strongly positively associated with pyrodiversity ($\beta_{\text{prop.burn}} = 2.5$; CI = 2.4, 2.5), with a negative quadratic term ($\beta_{\text{prop.burn}^2} = -0.78$; CI = $-0.80$, $-0.77$). These parameter estimates indicate a pyrodiversity peak when an average of 63% (CI = 61%, 65%) of a watershed has burned between 1985 and 2018 (electronic supplementary material, table S1; figure 4a). This apparent maximum equates to a 53-year fire rotation (CI = 51, 54 years), a measure of the time required to burn an area equivalent to the size of a landscape. In some cases, the combined direct and indirect effects on pyrodiversity are reinforcing (e.g. topography) while others dampen their ultimate influence (e.g. climate). For a given level of fire activity, pyrodiversity is negatively associated with CWD ($\beta_{\text{CWD}} = -0.048$; $-0.068$, $-0.029$) and AET ($\beta_{\text{AET}} = -0.019$; CI = $-0.035$, $-0.003$) with a positive interaction ($\beta_{\text{AET}*\text{CWD}} = 0.014$; CI = 0.001, 0.027) between the two climate variables. The combined marginal indirect and direct effects show CWD and AET interact to produce low pyrodiversity when watersheds lack an annual dry period but high pyrodiversity in productive areas coupled with dry periods (figure 4b). Similar to the burn activity model, elevation ($\beta_{\text{elev}} = 0.035$; CI = 0.013, 0.057) and topographic roughness ($\beta_{\text{rough}} = 0.016$; CI = 0.004, 0.028) are positively associated with pyrodiversity, with a likely slight negative interaction between the two ($\beta_{\text{elev}*\text{rough}} = -0.013$; CI = $-0.026$, 0.00). Consequently, pyrodiversity is maximized either at higher elevations or relatively low elevations with variable topography (figure 4c). When accounting for the level of burn activity, the direct effect of human population density on pyrodiversity is positive ($\beta_{\text{pop.den}} = 0.029$; CI = 0.017, 0.04) and proportion wilderness shows no clear direct effect

($\beta_{\text{wild}} = 0.022$; CI = $-0.022$, 0.066). Combined with a clearly positive indirect effect of proportion wilderness, and negative indirect effect of population density, watersheds in the designated wilderness have higher pyrodiversity on average, while more populated areas have marginally lower pyrodiversity (figure 4d,e).

## 4. Discussion

Pyrodiversity has received considerable attention in recent years as the inevitability of wildfire and its fundamental ecological role is increasingly recognized. While pyrodiversity clearly has appeal and applicability to many ecological disciplines, to date the concept remains nebulous with varied and often narrow definitions. Here, we present a generalizable functional diversity approach to quantifying pyrodiversity and tested its drivers across forested watersheds of the western United States. At the intermediate watershed scale, pyrodiversity was strongly but nonlinearly related to fire activity with an observed peak when approximately 63% of the flammable land area burned over the study period (equivalent to a 53-year fire rotation). Of the ultimate drivers tested, climate and proportion wilderness showed the strongest controls on pyrodiversity with productive but seasonally dry watersheds in wilderness areas most often characterized by variable fire histories. Areas with high topographic roughness or high elevation as well as areas with low human population density also tended to be more pyrodiverse. Correlations among individual pyrodiversity traits declined with the number of fires observed in a given watershed, suggesting previous use of a single fire regime trait (e.g. severity [13,14]) may be adequate for describing pyrodiversity following isolated fire events but is insufficient for characterizing landscapes with active fire regimes. A multi-dimensional approach supported by moderate- to high-resolution spatial data is likely necessary to capture the inherent complexity of fire across landscapes and bioregions.

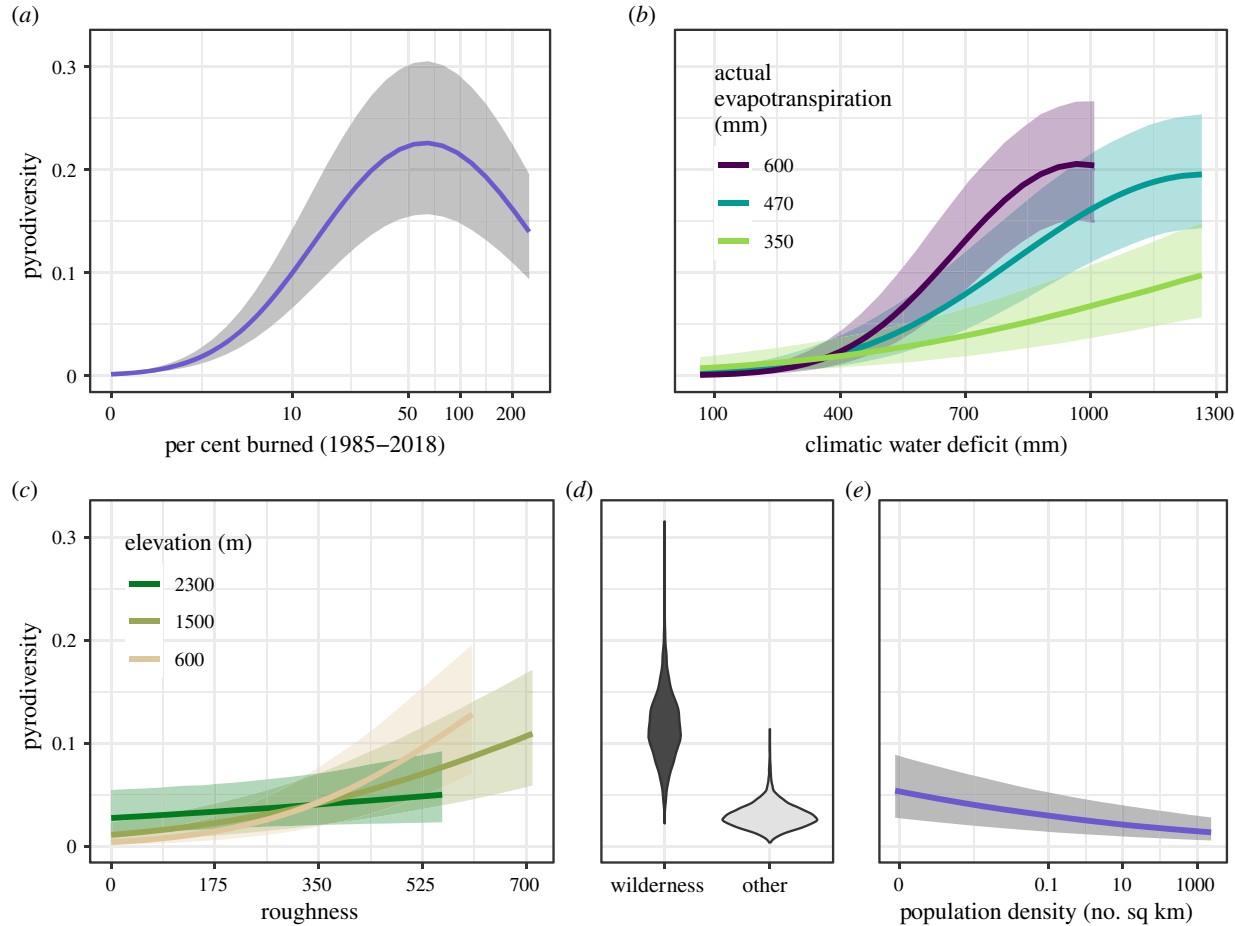

**Figure 4.** Drivers of watershed-scale pyrodiversity. (*a*) Cumulative per cent of flammable area burned from 1985 to 2018, (*b*) interacting climate effects of water deficit and actual evapotranspiration, (*c*) interacting topographic effects of roughness and elevation, (*d*) effect of wilderness designation and (*e*) effect of human population density. (*b*–*e*) Reflect both direct and burn activity-mediated effects. The effect of wilderness is modelled as a proportion of land area, but binary marginal effects are presented here for simplicity. Pyrodiversity is defined as the multi-variate dispersion of fire frequency, severity, seasonality and patch size. (Online version in colour.)

## (a) Quantifying pyrodiversity

An ideal metric of pyrodiversity considers multiple fire regime traits in a functional diversity framework, allows for flexible treatment of the invisible mosaic, is applicable at both fine scales (i.e. evaluates within-fire variation) and across broad spatio-temporal extents, and is easily reproducible to allow comparison among regions. We build upon previous pyro-diversity assessments (electronic supplementary material, table S2) to develop such a metric. In particular, we build on Hempson *et al.*'s [18] multi-trait index approach by quantifying within-fire variation, incorporating flexible order-weighting of fire events and adopting FDis as our metric of pyrodiversity. Where the ecological process or pattern of interest occurs at fine scales quantification of within-fire variation (e.g. using 30 m resolution data) is necessary. Further, FDis appears more useful than functional richness (multi-variate range) when considering traits such as burn severity where the full range of variation is represented in most burned landscapes (i.e. 0–100% vegetation mortality; electronic supplementary material). The principle disadvantage of our approach is that currently remotely sensed estimates of burn severity are most reliable in areas dominated by forests [10], making assessments in other biomes more tenuous. Regardless of the approach employed, our ability to quantify historical fire patterns is challenged by the temporal availability of data, which restricts such assessments to recent pyrodiversity only. Pyrodiversity may be

particularly difficult to quantify with precision within fire regimes with long rotation times. Assessments focused on areas with restored fire regimes or simulation studies may be necessary to estimate historical levels of pyrodiversity. Landsat imagery allows for assessments starting in 1984, while the coarser-scale MODIS imagery is possible beginning in 2000. As methods for estimating burn severity outside of forests are improved and where fire frequency and seasonality data are available for a longer period [12] these limitations can be partially alleviated.

## (b) Drivers of pyrodiversity

Climate exerts strong controls on biome distribution and fire regimes globally, while topography is often omitted from or considered less important in assessments at the fire regime level [19,34] (but see [21]). Here, we establish that climatic control extends to variation in current fire patterns both directly and indirectly as mediated by burn activity (figure 4*b*). Relative to climate, we found elevation and topo-graphic roughness to have small but meaningful effects on burn activity and pyrodiversity (figure 4*c*; electronic supplementary material, table S1). Hempson *et al.* [18] found a negative relationship between pyrodiversity and precipitation with a pyrodiversity peak in dry areas of Africa, but no discernible effect of topographic roughness. This observed

relationship with precipitation is consistent with our finding that pyrodiversity increases with CWD but is somewhat at odds with our finding of a positive relationship with AET, which is related to precipitation. Together these assessments indicate pyrodiversity is dependent both on the production of vegetative biomass and its seasonal availability to burn as fuel. Topographic roughness may be important in supporting intra-fire variability if rapid changes in terrain moderate fire behaviour and break up patches of fire severity [21]. However, the topography may exert little control on the variability of fire-level metrics such as fire size and maximum burn intensity in some areas [18].

We interpret the negative relationship between human population density and pyrodiversity to reflect highly successful fire exclusion and suppression efforts across much of North America [24]. Changes in vegetative structure and fire patterns attributable to fire suppression have already been documented in fire-adapted ecosystems [23,29,35], and these findings indicate pyrodiversity is almost certainly lower in such systems than historic levels [36]. The strong positive effect of proportion wilderness likely reflects fire policies of many US wilderness areas that strive to restore pre-suppression era fire regimes [37]. Wilderness areas that explicitly allow lightning-caused wildfires to be used for resource objectives [38] appear to contain greater levels of pyrodiversity. However, the benefit of wilderness is likely highly context dependent. Some of the most pyrodiverse areas in the western United States fall within wilderness areas such as Yosemite National Park [39], Frank Church–River of No Return Wilderness, Bob Marshall Wilderness and the Gila Wilderness [40], but not in the wilderness areas of Olympic National Park characterized by a very wet climate. Interestingly, Sequoia–Kings Canyon National Park in the southern Sierra Nevada of California was an early pioneer in the use of both prescribed and managed natural fire [38,41] but does not appear particularly pyrodiverse, while an area just to its south (Kern Plateau, Sequoia National Forest) does (figure 2).

The full nature of human influence on pyrodiversity is likely more complex than can be captured by the necessarily coarse measures of population density and wilderness designation. At sub-watershed scales, the use of prescribed and cultural burning are likely important contributors to pyrodiversity in some areas [42,43]. Tribal burning in California, for example, serves an array of cultural purposes and creates diverse habitat mosaics that sustained meadows, woodlands, wetlands, coastal prairies and grasslands [42,44]. Many Tribes used a system of patch burning that manipulated vegetation at fine spatial scale to meet their management objectives. How these cultural fire regimes impact pyrodiversity deserves continued evaluation where fire histories exist at finer scales than captured by the national MTBS dataset.

## (c) Changing fire regimes

Humans have altered fire regimes directly through management and indirectly by altering the Earth's climate, and such shifts are almost certainly also changing pyrodiversity. Perhaps the most clear effects of altered fire regimes on pyrodiversity are seen where fire exclusion and suppression policies have dramatically reduced burned area and shifted fire severity patterns in fire-adapted forests [45]. Conversely, climate change is increasing fire activity by lengthening fire seasons and increasing water deficits [36,46–48]. Given the strong link between CWD, fire activity and pyrodiversity, these changes may increase pyrodiversity in the short-term where deficits of fire activity currently exist but could result in lower levels of pyrodiversity for areas with high levels of contemporary burn activity (figure 4a). Additionally, in many areas, larger fires are increasingly accompanied by ever larger and simpler shaped patches of high-severity effects [49,50], which could result in lower pyrodiversity at fine scales. Where the frequency of high-severity fire exceeds the natural range of variation of an ecosystem, higher rates of type-conversion (e.g. from forests to shrubland) may occur [51,52]. This may be particularly problematic in dry areas where a further increase in water deficit can lead to a consistent loss in productivity [36] or when wildfires interact with other climate-exacerbated disturbances such as periodic drought and beetle infestations [53]. Ultimately, climate-related shifts in pyrodiversity are likely to be uneven across the western United States and globally. How these changes impact biodiversity and ecosystem process may depend on whether emerging pyrodiversity patterns result in a dramatic departure from historic fire regimes.

## (d) Improving our understanding of pyrodiversity's ecological role

Variation in spatio-temporal fire patterns plays an important role in various ecological processes including resilience to future disturbance and hydraulic function [54,55], but to date, the principal applications of the pyrodiversity concept have focused on its relationship with biodiversity. In the three decades since Martin & Sapsis [8] first articulated the hypothesis that pyrodiversity begets biodiversity, an increasing number of studies have provided evidence to support their theory [12–14,17,56,57], while others have found the relationship to be weak or non-existent [57–60]. These occasionally conflicting findings as well as our results showing high variation in pyrodiversity across ecosystems indicate the functional relationship between pyrodiversity and biodiversity may not be absolute but rather is limited or context dependent. For example, we observed a maximum pyrodiversity among watersheds with an approximate 53-year fire rotation. This rate of fire activity and pyrodiversity is unlikely to optimize biodiversity across all ecosystems with highly varied historical relationships with wildfire, including forests adapted to frequent surface fire and others adapted to infrequent crown fire.

We hypothesize that constraints to the pyrodiversity–biodiversity relationship are related to an ecosystem's historical fire regime and that on average biodiversity may maximize at levels of the pyrodiversity characteristic of the conditions under which ecological communities assembled. Further, the relative importance of individual pyrodiversity traits and the underlying mechanisms by which pyrodiversity affects biodiversity [30] may also vary among historic fire regimes. This hypothesis leads to expected and testable functional forms under different conditions. Peak biodiversity may occur at moderate to high levels of pyrodiversity for fire regimes characterized by frequent fire and relatively small but variable high-severity patch sizes, such as those found in the semi-dry forests of North America (figure 5a). In less active fire regimes such as wet temperate forests, the biodiversity peak may occur at lower levels of pyrodiversity

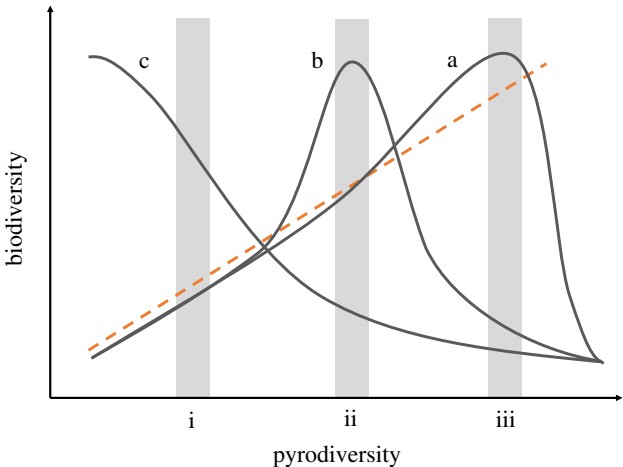

**Figure 5.** Theoretical functional relationships between pyrodiversity and biodiversity. A positive and absolute relationship is shown as an orange dashed line and solid lines represent example ecosystems where the relationship is limited by the historic fire regime, where peak biodiversity may occur at high (a), moderate (b) or low (c) levels of pyrodiversity. Our ability to perceive the full functional form is limited by the environmental space sampled and may appear linear or non-existent when the full range of pyrodiversity is not observed (i–iii). (Online version in colour.)

either because fire-adapted species have been filtered from the regional species pool and/or fire-adaptive traits have not evolved *in situ* [61]. Ecosystems with little variation in burn severity such as savannahs may see an analogous mid-pyrodiversity peak, above which more severe fires threaten to convert the system to grassland (figure 5b). The threat of tipping points or type-conversions may be especially acute in ecosystems like tropical rainforest which have little to no history of lightning-ignited fire and to which native species are poorly adapted [62]. Where fire activity and pyrodiversity increase in these ecosystems the biodiversity response may be predominantly negative (figure 5c). The theoretical dependence of the pyrodiversity–biodiversity relationship on historic fire regimes is supported by Miller & Safford [61] who provide evidence that plant biodiversity is maximized where average burn severities match the predominant historical disturbance regime of an ecosystem. Alternatively, He *et al.* [4] proposed the association is constrained by species : area relationships and that at very high levels of pyrodiversity declining patch sizes limit the number of species present.

In addition to uncertainties surrounding the mechanisms of the pyrodiversity–biodiversity relationship, perceiving the full pyrodiversity–biodiversity functional form is dependent on the range of pyrodiversity observed. Partially observed relationships could be attributed to limited sampling effort or modern shifts in fire regimes away from historic conditions. For example, where fire activity has been artificially reduced, pyrodiversity may be lower than the biodiversity optimum across a study region and biodiversity would appear to increase with pyrodiversity absolutely ([14]; figure 5i). Indeed, Martin & Sapsis [8] developed their original theory in the context of extensive fire suppression in the mixed-conifer forests of California, where the detrimental effects of a dearth of pyrodiversity was clear. Observational studies conducted within landscapes containing a wide range of fire patterns, comprehensive fire history datasets (including small fires) and a generalizable method of quantifying pyrodiversity may be necessary to fully resolve the phenomenon's relationship with biodiversity and ecosystem process across ecosystems.

## 5. Conclusion

We developed a generalizable trait-based approach and provide reproducible code for quantifying pyrodiversity at regional to local scales. This method builds on previous efforts to quantify pyrodiversity by (i) using a functional diversity framework that captures multi-dimensional dispersion of pyrodiversity traits; (ii) incorporating Landsat imagery and Google Earth Engine to measure intra-fire variation anywhere validated severity models exist and (iii) allowing flexible weighting of individual fire traits and the relative importance of the visible/invisible mosaic. While we demonstrate its utility at the regional scale, the 30 m resolution of the underlying data also allow calculation of pyrodiversity at scales relevant to point or plot-based ecological sampling methods. This approach, along with an improved understanding of the ultimate drivers of pyrodiversity provides opportunities to more consistently and comprehensively test the influence of pyrodiversity on biodiversity and other ecosystem processes. Doing so across regions, management approaches and ecological communities will increase our ability to manage fire, maintain ecosystem function and conserve biodiversity as fire regimes continue to shift with accelerating global change.

Data accessibility. Data supporting results are publicly available at https://figshare.com/articles/Pyrodiversity_westCONUS/12478832, and code used to generate data layers and analysis is available at https://github.com/zacksteel/pyrodiversity.

Authors' contributions. Z.S., B.C. and S.S. conceived the study. Z.S. performed data collection and analysis, with input from B.C., S.S. and D.S. Z.S. wrote the first draft of the manuscript, and all authors contributed substantially to revisions.

Competing interests. The authors have no competing interests.

Funding. We received no funding for this study.

Acknowledgements. We thank Sean Parks and Lisa Holsinger for assistance with data generation using Google Earth Engine. Current and former members of the Stephens, and Safford labs as well as two anonymous reviewers were helpful when refining early manuscript drafts.

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
