## [Peer Review File · Proceedings of the Royal Society B: Biological Sciences]

Review History

RSPB-2020-1670.R0 (Original submission)

Review form: Reviewer 1

Recommendation

Major revision is needed (please make suggestions in comments)

Scientific importance: Is the manuscript an original and important contribution to its field?

Good

General interest: Is the paper of sufficient general interest?

Good

Quality of the paper: Is the overall quality of the paper suitable?

Excellent

Is the length of the paper justified?

Yes

Should the paper be seen by a specialist statistical reviewer?

No

Do you have any concerns about statistical analyses in this paper? If so, please specify them explicitly in your report.

No

It is a condition of publication that authors make their supporting data, code and materials available - either as supplementary material or hosted in an external repository. Please rate, if applicable, the supporting data on the following criteria.

Is it accessible?

Yes

Is it clear?

Yes

Is it adequate?

Yes

Do you have any ethical concerns with this paper?

No

Comments to the Author

Summary:

In this paper, the authors present a well thought out, flexible, and reproducible method for quantifying pyrodiversity in the western US based on four fire regime traits. The authors further quantify the relationship between direct and indirect (mediated by burn activity) drivers of pyrodiversity, accounting for climate, topography, and human influence. Finally, the authors present a hypothesis about the relationship between pyrodiversity and biodiversity based on a review of previous studies. This paper is well written, the statistical methods are well-described and seem appropriate, and the focus on reproducibility and flexibility makes this a valuable contribution for future studies. The methodological approach and results are well supported by figures and tables. However, while the hypothesized relationship between pyrodiversity and biodiversity is interesting and of broad appeal, it does not clearly follow from the results of this study and instead seems “tacked on”. In addition, this paper would be strengthened by a quantitative comparison of this new method of calculating pyrodiversity with previous methods, which would enable an assessment of how and where it is superior to these previous methods.

General comments:

(1) The relationship between pyrodiversity and biodiversity is a central focus of the framing of this paper in the introduction (e.g., lines 64-77, 97-98) and of an entire section in the discussion proposing a new hypothesis (lines 416-490). While the hypothesis is interesting and compelling, it is not clear how it follows from the analyses and results presented in this paper. One suggestion is for the authors to conduct an additional analysis asking how their newly calculated pyrodiversity metric is related to some metric of biodiversity. If this analysis is performed using the same biodiversity data and at the same scale as some of the previous studies referenced, this could also enable comparison between different methods of quantifying pyrodiversity (which would also address my second general comment).

(2) The authors do an excellent job of pointing out limitations of previous methods of calculating pyrodiversity and articulating how this new method should represent an improvement. These assertions would be strengthened by a comparison and assessment of different methods of calculating pyrodiversity. Does this new method indeed improve our ability to quantify and understand pyrodiversity? Is this method of calculating pyrodiversity more consistent with our ecological understanding of differences among watersheds? How could this be assessed?

(3) I suggest adding a limitations paragraph to the discussion – some limitations of this approach are discussed (e.g., lines 73-75, 405-414), but they are scattered and incomplete. The primary limitation appears to be the duration of MTBS availability, meaning that the “invisible mosaic” is not fully captured. How does this affect estimated pyrodiversity? This seems like it would be particularly problematic for areas that burned very little during this time period or for areas where fire rotations tend to be very long. This also makes it challenging to assess historical pyrodiversity using consistent methods, which seems like it would be particularly important if emulating historical pyrodiversity is the hypothesized management goal for maximizing biodiversity.

Specific comments:

Line 58. I found “expanded scrutiny” to be an odd choice of words here. Perhaps “expanded use”?

Lines 73-75, 525-527. It is unclear whether the ability to quantify the “invisible mosaic” is proposed as a strength (due to including decay rate) or a weakness (due to only ~35 years of MTBS data) of this study. Perhaps both?

Lines 124-202. I suggest describing the traits first and pyrodiversity calculation second.

Figure 1. Excellent and helpful figure.

Lines 163-164. What fire return interval value is assigned to pixels with no fire in the MTBS record?

Line 170. Typo, “,29”

Line 219. The ability to quantify both direct and indirect effects is very compelling and leads to some interesting results and discussion.

Lines 292-303. I recommend noting that predictors are standardized somewhere in the main text (in addition to in the supplement).

Line 399. Typo “wildness”

Review form: Reviewer 2

Recommendation

Major revision is needed (please make suggestions in comments)

Scientific importance: Is the manuscript an original and important contribution to its field?

Acceptable

General interest: Is the paper of sufficient general interest?

Acceptable

Quality of the paper: Is the overall quality of the paper suitable?

Acceptable

Is the length of the paper justified?

No

Should the paper be seen by a specialist statistical reviewer?

No

Do you have any concerns about statistical analyses in this paper? If so, please specify them explicitly in your report.

No

It is a condition of publication that authors make their supporting data, code and materials available - either as supplementary material or hosted in an external repository. Please rate, if applicable, the supporting data on the following criteria.

Is it accessible?

Yes

Is it clear?

Yes

Is it adequate?

Yes

Do you have any ethical concerns with this paper?

No

Comments to the Author

This manuscript proposes a new, general approach to quantifying pyrodiversity – and assesses how this index varies in response to environmental and anthropogenic factors. While the use of functional dispersion offers much potential in quantifying pyrodiversity in a generalizable manner, some of the decisions regarding how fire traits are calculated and what sampling units are used are somewhat debatable. I also feel that more could be done to integrate this research into the existing literature. In particular, Hempson et al. 2018 *Ecography* essentially provides the roadmap for this study, with Beale et al. 2018 *Ecology Letters* being the next stop where a general metric of pyrodiversity is used to assess consequences for biodiversity – the apparent premise for this work. That the approach of Hempson et al. and that proposed here could easily each be applied to the respective datasets seems to have been overlooked – it would be interesting to see a more critical assessment of the differences and strengths of each approach.

Specific comments

Line 93-95: It is not clear why the authors consider Hempson et al.'s approach to quantifying pyrodiversity as being restricted to coarse-scales – they calculate pyrodiversity for a range of spatial grains from 15 to 120 m grids, and indeed analyse whether pyrodiversity is dependent on spatial grain. Given the availability of fire trait data, there is no limit to the spatial grain at which their approach could be applied. Similarly, it is not clear what is meant by the lack of ability to capture within-fire traits i.e. 'such as variation in burn severity and spatial pattern (i.e. patch size)' – there would seem to be no reason why Hempson et al.'s method could not be applied to the 30 m grain fire trait data in this study. Given that your aim seems to be much the same as that of Hempson et al. (i.e. Hempson et al. abstract: "We present the first generalizable method to quantify pyrodiversity, and use it to address the fundamental questions of what drives pyrodiversity, which fire attributes constrain pyrodiversity under different conditions, and whether pyrodiversity is spatial grain-dependent."), I think you need to do more to justify what a new approach could offer.

Line 134-135: 'unique combinations of fire regime traits (fire histories) are considered individual species' – please clarify what the sample unit is. Are the units under assessment individual fires, or are they grid cells in a gridded landscape?

Lines 133-136: This could be clarified – it goes from species to community and back to species. Am I correct in understanding that: 1) the landscape = a community of pixels (species), and 2) if two pixels share the same fire traits (histories), they are considered to be the same species? What size pixel is used [I see this is clarified later]? How does pixel size – through its effect on the number of weighted-distance-to-centroid values to be averaged (i.e. number of pixels in the community/landscape) – influence the pyrodiversity/FDis value?

From what I can tell, a single position in trait space is calculated for each pixel (species) – as such, there is a single value for each fire trait for each pixel – if so, this means that any variability in that fire trait within the pixel gets summarised to a single value. Accordingly, your pyrodiversity index is very likely to be dependent on pixel size, as a consequence of how much fire trait variation gets summarised to a single pixel-value as pixel size changes. Is this the case?

By analogy – it's like gridding a landscape, recording what trees (fires) you see in each pixel, and averaging/summarising their height, canopy diameter, bark thickness and SLA (four fire traits) – and then calculating FDis using the pixels/grid cells in the landscape. FDis will be robust to the number of pixels ('species richness') in the sense that distance to centroid will be a weighted-average, but the position of that pixel in trait space will be dependent on what trees occur in each pixel and hence the distribution of trait values being summarised. Starting from a single pixel representing the whole landscape, I'd expect the FDis value to increase steadily as pixel size gets smaller and spatial variation in tree communities gets better represented, and to then start to stabilise – but I'd expect this stabilisation to occur sooner in a landscape with low tree diversity and spatial turnover, but much later (i.e. smaller pixel size) for a hyper-diverse region with a mosaic of different vegetation types.

Can this approach be generalised to contexts where the base data are at spatial grains considerably larger than the 30 m pixels in this study?

Line 136: Presumably no pixels shared exactly the same histories?

Line 144-146: This is the approach used by Hempson et al. [28] – note that they account for some of the effect of outliers by using a non-parametric bootstrap to account for the effect of the number of fires on the volume of the minimum convex hull.

How does your method control for the positive effect of the number of fires on pyrodiversity?

Line 146-148: Note that trait weighting is certainly not unique to FDis – Hempson et al. suggest that future work considers the merits of weighting traits differently while using their method (pg. 896), but neither their study nor yours actually implements differential weighting of traits.

Fig. 1B & C – these are not particularly informative – one cannot see the full trait surfaces, nor the pixel sizes – and the indirect effects of the three factors you choose to illustrate in C are implied by the arrows going via fire activity.

Line 172: Am I correct in understanding that you used the first and final year of the data availability period to calculate fire return interval? As in, if it burned once during the ~35 year period (e.g. year 10), the fire return intervals would be 10 years (10-start) and 25 years (end-10)? Are you recommending this as a general method, including for areas where fires are rare?

Line 179-181: Does this not confound burn severity with fire size, to some extent?

Line 186-194: This is a rather surprising assumption to be making when looking to formulate a generalised measure of pyrodiversity – what if 'biodiversity' does not respond by being "most sensitive to recent events but previous fires (the "invisible mosaic") maintain some influence over landscape pattern and process"? In many fire prone systems, occasional long fire intervals may allow for woody plants to escape the fire-trap, or occasional high intensity/late dry season fires may dramatically restructure vegetation and associated biota – the most consequential events are

not necessarily the most recent, and different taxa will respond on vastly different timescales. Furthermore, if a general researcher was interested in pyrodiversity for reasons other than effects on biodiversity (e.g. how does topography shape pyrodiversity), why should biodiversity considerations be built into the metric? When looking to quantify the variability in fire properties within a region, why should the most recent fires be accorded more weight?

Line 212-213: Please clarify how trait correlations were made for watersheds with zero fires – or else clarify what is meant by “Specifically, correlations were made among traits for all study watersheds with minimum number of fires ranging from zero to fifteen”

Line 215-250: I’m not quite sure that I follow the reasoning behind including proportion burned area (i.e. ‘fire activity’) in these models, particularly as it is inherent to at least two of the traits used to quantify pyrodiversity (i.e. fire frequency and patch size). Basically, pyrodiversity should be a measure of how variable fires are within a region – having many large fires is less pyrodiverse than a mix of small and large – having many short fire intervals is less pyrodiverse than a mix of short and long – with the former in each case likely to increase the proportion burned area i.e. ‘burn activity’.

Line 358-362: What do you mean by an isolated fire event? If pyrodiversity is conceptualised as the level of dispersion in multi-trait space (as you seem to quantify it) – then how can a single fire trait be used to adequately quantify pyrodiversity? Given the goal of developing a general metric of pyrodiversity, when would you suggest using different numbers of traits?

Decision letter (RSPB-2020-1670.R0)

24-Aug-2020

Dear Dr Steel:

I am writing to inform you that your manuscript RSPB-2020-1670 entitled "Quantifying pyrodiversity and its drivers" has, in its current form, been rejected for publication in Proceedings B.

This action has been taken on the advice of referees, who have recommended that substantial revisions are necessary. With this in mind we would be happy to consider a resubmission, provided the comments of the referees are fully addressed. However please note that this is not a provisional acceptance.

- 1) A ‘response to referees’ document including details of how you have responded to the comments, and the adjustments you have made.
- 2) A clean copy of the manuscript and one with 'tracked changes' indicating your 'response to referees' comments document.
- 3) Line numbers in your main document.

4) Data - please see our policies on data sharing to ensure that you are complying (<https://royalsociety.org/journals/authors/author-guidelines/#data>).

Sincerely,
Dr Maurine Neiman
mailto: proceedingsb@royalsociety.org

Associate Editor
Board Member: 1
Comments to Author:

The authors of this manuscript proposed a new index measuring pyrodiversity based on fire regime traits. This index is based on the functional diversity definition and is flexible and reproducible in quantifying pyrodiversity. Using this new index, the authors demonstrated the geographical patterns in pyrodiversity in western US, and then explored the influences of fire regimes, climate, topographic roughness, wilderness and human population density on pyrodiversity. The authors also discussed the possible relationships between biodiversity and pyrodiversity. Two reviewers reviewed this manuscript. Although both of them recognized the merits of this manuscript, they also provided many critical comments on the analyses, interpretation of the results, and the presentation of the index. I think these comments are very useful for the authors to improve the current manuscript. I would like to reinforce a couple of points raised by the reviewers.

First, as pointed out by both reviewers, the authors did not put their study in more clear context. Both reviewers suggested that the authors should more clearly compare their new index with previous ones. Reviewer #2 also pointed that the current index is relatively similar to the index proposed by Hempson et al. 2018 in *Ecography*, and used by Beale et al. 2018 in *Ecology Letters*. I agree with both reviewers. I think the authors should carefully compare their index with previous ones in terms of the advantages and disadvantages of different indices, and their outputs in ecological research. Moreover, I think the authors should also try to make the description about the fire regime traits clearer.

Second, the authors could evaluate the relationship between pyrodiversity and biodiversity using both the new index and the previous ones (e.g. the one proposed by Hempson et al. 2018), and compare the results based on different indices.

Third, I think the authors should better present their study in the context of previous studies in the same field. Moreover, writing of the manuscript would benefit from a more careful language proofing.

Reviewer(s)' Comments to Author:

Referee: 1

Comments to the Author(s)

Summary:

In this paper, the authors present a well thought out, flexible, and reproducible method for quantifying pyrodiversity in the western US based on four fire regime traits. The authors further quantify the relationship between direct and indirect (mediated by burn activity) drivers of pyrodiversity, accounting for climate, topography, and human influence. Finally, the authors present a hypothesis about the relationship between pyrodiversity and biodiversity based on a review of previous studies. This paper is well written, the statistical methods are well-described and seem appropriate, and the focus on reproducibility and flexibility makes this a valuable contribution for future studies. The methodological approach and results are well supported by figures and tables. However, while the hypothesized relationship between pyrodiversity and

biodiversity is interesting and of broad appeal, it does not clearly follow from the results of this study and instead seems “tacked on”. In addition, this paper would be strengthened by a quantitative comparison of this new method of calculating pyrodiversity with previous methods, which would enable an assessment of how and where it is superior to these previous methods.

General comments:

(1) The relationship between pyrodiversity and biodiversity is a central focus of the framing of this paper in the introduction (e.g., lines 64-77, 97-98) and of an entire section in the discussion proposing a new hypothesis (lines 416-490). While the hypothesis is interesting and compelling, it is not clear how it follows from the analyses and results presented in this paper. One suggestion is for the authors to conduct an additional analysis asking how their newly calculated pyrodiversity metric is related to some metric of biodiversity. If this analysis is performed using the same biodiversity data and at the same scale as some of the previous studies referenced, this could also enable comparison between different methods of quantifying pyrodiversity (which would also address my second general comment).

(2) The authors do an excellent job of pointing out limitations of previous methods of calculating pyrodiversity and articulating how this new method should represent an improvement. These assertions would be strengthened by a comparison and assessment of different methods of calculating pyrodiversity. Does this new method indeed improve our ability to quantify and understand pyrodiversity? Is this method of calculating pyrodiversity more consistent with our ecological understanding of differences among watersheds? How could this be assessed?

(3) I suggest adding a limitations paragraph to the discussion – some limitations of this approach are discussed (e.g., lines 73-75, 405-414), but they are scattered and incomplete. The primary limitation appears to be the duration of MTBS availability, meaning that the “invisible mosaic” is not fully captured. How does this affect estimated pyrodiversity? This seems like it would be particularly problematic for areas that burned very little during this time period or for areas where fire rotations tend to be very long. This also makes it challenging to assess historical pyrodiversity using consistent methods, which seems like it would be particularly important if emulating historical pyrodiversity is the hypothesized management goal for maximizing biodiversity.

Specific comments:

Line 58. I found “expanded scrutiny” to be an odd choice of words here. Perhaps “expanded use”?

Lines 73-75, 525-527. It is unclear whether the ability to quantify the “invisible mosaic” is proposed as a strength (due to including decay rate) or a weakness (due to only ~35 years of MTBS data) of this study. Perhaps both?

Lines 124-202. I suggest describing the traits first and pyrodiversity calculation second.

Figure 1. Excellent and helpful figure.

Lines 163-164. What fire return interval value is assigned to pixels with no fire in the MTBS record?

Line 170. Typo, “,29”

Line 219. The ability to quantify both direct and indirect effects is very compelling and leads to some interesting results and discussion.

Lines 292-303. I recommend noting that predictors are standardized somewhere in the main text (in addition to in the supplement).

Line 399. Typo “wildness”

Referee: 2

Comments to the Author(s)

This manuscript proposes a new, general approach to quantifying pyrodiversity – and assesses how this index varies in response to environmental and anthropogenic factors. While the use of functional dispersion offers much potential in quantifying pyrodiversity in a generalizable manner, some of the decisions regarding how fire traits are calculated and what sampling units are used are somewhat debatable. I also feel that more could be done to integrate this research into the existing literature. In particular, Hempson et al. 2018 *Ecography* essentially provides the roadmap for this study, with Beale et al. 2018 *Ecology Letters* being the next stop where a general metric of pyrodiversity is used to assess consequences for biodiversity – the apparent premise for this work. That the approach of Hempson et al. and that proposed here could easily each be applied to the respective datasets seems to have been overlooked – it would be interesting to see a more critical assessment of the differences and strengths of each approach.

Specific comments

Line 93-95: It is not clear why the authors consider Hempson et al.’s approach to quantifying pyrodiversity as being restricted to coarse-scales – they calculate pyrodiversity for a range of spatial grains from 15 to 120 m grids, and indeed analyse whether pyrodiversity is dependent on spatial grain. Given the availability of fire trait data, there is no limit to the spatial grain at which their approach could be applied. Similarly, it is not clear what is meant by the lack of ability to capture within-fire traits i.e. ‘such as variation in burn severity and spatial pattern (i.e. patch size)’ – there would seem to be no reason why Hempson et al.’s method could not be applied to the 30 m grain fire trait data in this study. Given that your aim seems to be much the same as that of Hempson et al. (i.e. Hempson et al. abstract: “We present the first generalizable method to quantify pyrodiversity, and use it to address the fundamental questions of what drives pyrodiversity, which fire attributes constrain pyrodiversity under different conditions, and whether pyrodiversity is spatial grain-dependent.”), I think you need to do more to justify what a new approach could offer.

Line 134-135: ‘unique combinations of fire regime traits (fire histories) are considered individual species’ – please clarify what the sample unit is. Are the units under assessment individual fires, or are they grid cells in a gridded landscape?

Lines 133-136: This could be clarified – it goes from species to community and back to species. Am I correct in understanding that: 1) the landscape = a community of pixels (species), and 2) if two pixels share the same fire traits (histories), they are considered to be the same species?

What size pixel is used [I see this is clarified later]? How does pixel size – through its effect on the number of weighted-distance-to-centroid values to be averaged (i.e. number of pixels in the community/landscape) – influence the pyrodiversity/FDis value?

From what I can tell, a single position in trait space is calculated for each pixel (species) – as such, there is a single value for each fire trait for each pixel – if so, this means that any variability in that fire trait within the pixel gets summarised to a single value. Accordingly, your pyrodiversity index is very likely to be dependent on pixel size, as a consequence of how much fire trait variation gets summarised to a single pixel-value as pixel size changes. Is this the case?

By analogy – it’s like gridding a landscape, recording what trees (fires) you see in each pixel, and averaging/summarising their height, canopy diameter, bark thickness and SLA (four fire traits) –

and then calculating FDis using the pixels/grid cells in the landscape. FDis will be robust to the number of pixels ('species richness') in the sense that distance to centroid will be a weighted-average, but the position of that pixel in trait space will be dependent on what trees occur in each pixel and hence the distribution of trait values being summarised. Starting from a single pixel representing the whole landscape, I'd expect the FDis value to increase steadily as pixel size gets smaller and spatial variation in tree communities gets better represented, and to then start to stabilise – but I'd expect this stabilisation to occur sooner in a landscape with low tree diversity and spatial turnover, but much later (i.e. smaller pixel size) for a hyper-diverse region with a mosaic of different vegetation types.

Can this approach be generalised to contexts where the base data are at spatial grains considerably larger than the 30 m pixels in this study?

Line 136: Presumably no pixels shared exactly the same histories?

Line 144-146: This is the approach used by Hempson et al. [28] – note that they account for some of the effect of outliers by using a non-parametric bootstrap to account for the effect of the number of fires on the volume of the minimum convex hull.

How does your method control for the positive effect of the number of fires on pyrodiversity?

Line 146-148: Note that trait weighting is certainly not unique to FDis – Hempson et al. suggest that future work considers the merits of weighting traits differently while using their method (pg. 896), but neither their study nor yours actually implements differential weighting of traits.

Fig. 1B & C – these are not particularly informative – one cannot see the full trait surfaces, nor the pixel sizes – and the indirect effects of the three factors you choose to illustrate in C are implied by the arrows going via fire activity.

Line 172: Am I correct in understanding that you used the first and final year of the data availability period to calculate fire return interval? As in, if it burned once during the ~35 year period (e.g. year 10), the fire return intervals would be 10 years (10-start) and 25 years (end-10)? Are you recommending this as a general method, including for areas where fires are rare?

Line 179-181: Does this not confound burn severity with fire size, to some extent?

Line 186-194: This is a rather surprising assumption to be making when looking to formulate a generalised measure of pyrodiversity – what if 'biodiversity' does not respond by being "most sensitive to recent events but previous fires (the "invisible mosaic") maintain some influence over landscape pattern and process"? In many fire prone systems, occasional long fire intervals may allow for woody plants to escape the fire-trap, or occasional high intensity/late dry season fires may dramatically restructure vegetation and associated biota – the most consequential events are not necessarily the most recent, and different taxa will respond on vastly different timescales. Furthermore, if a general researcher was interested in pyrodiversity for reasons other than effects on biodiversity (e.g. how does topography shape pyrodiversity), why should biodiversity considerations be built into the metric? When looking to quantify the variability in fire properties within a region, why should the most recent fires be accorded more weight?

Line 212-213: Please clarify how trait correlations were made for watersheds with zero fires – or else clarify what is meant by "Specifically, correlations were made among traits for all study watersheds with minimum number of fires ranging from zero to fifteen"

Line 215-250: I'm not quite sure that I follow the reasoning behind including proportion burned area (i.e. 'fire activity') in these models, particularly as it is inherent to at least two of the traits used to quantify pyrodiversity (i.e. fire frequency and patch size). Basically, pyrodiversity should be a measure of how variable fires are within a region – having many large fires is less

pyrodiverse than a mix of small and large – having many short fire intervals is less pyrodiverse than a mix of short and long – with the former in each case likely to increase the proportion burned area i.e. ‘burn activity’.

Line 358-362: What do you mean by an isolated fire event? If pyrodiversity is conceptualised as the level of dispersion in multi-trait space (as you seem to quantify it) – then how can a single fire trait be used to adequately quantify pyrodiversity? Given the goal of developing a general metric of pyrodiversity, when would you suggest using different numbers of traits?

Author's Response to Decision Letter for (RSPB-2020-1670.R0)

See Appendix A.

RSPB-2020-3202.R0

Review form: Reviewer 1

Recommendation

Accept with minor revision (please list in comments)

Scientific importance: Is the manuscript an original and important contribution to its field?

Good

General interest: Is the paper of sufficient general interest?

Excellent

Quality of the paper: Is the overall quality of the paper suitable?

Excellent

Is the length of the paper justified?

Yes

Should the paper be seen by a specialist statistical reviewer?

No

Do you have any concerns about statistical analyses in this paper? If so, please specify them explicitly in your report.

No

It is a condition of publication that authors make their supporting data, code and materials available - either as supplementary material or hosted in an external repository. Please rate, if applicable, the supporting data on the following criteria.

Is it accessible?

Yes

Is it clear?

Yes

Is it adequate?

Yes

Do you have any ethical concerns with this paper?

No

Comments to the Author

General comments

I provided reviewer comments on a previous draft of this manuscript. The authors have adequately addressed comments from the editor and reviewers, including adding a thorough comparison of multiple methods of calculating pyrodiversity in the main text and supplementary material. The authors have also provided an understandable rationale for not including an analysis of the relationship between pyrodiversity and biodiversity within this manuscript. This paper is well written, introduces a compelling and reproducible method for calculating pyrodiversity, and sets the stage for additional exploration of the relationship between pyrodiversity and ecological patterns and processes.

Specific comments

Line 67. Typo, excess “;”

Line 375. Typo, “evaluates”

Line 381. Typo, “complementary”

Review form: Reviewer 2

Recommendation

Major revision is needed (please make suggestions in comments)

Scientific importance: Is the manuscript an original and important contribution to its field?

Acceptable

General interest: Is the paper of sufficient general interest?

Good

Quality of the paper: Is the overall quality of the paper suitable?

Acceptable

Is the length of the paper justified?

Yes

Should the paper be seen by a specialist statistical reviewer?

No

Do you have any concerns about statistical analyses in this paper? If so, please specify them explicitly in your report.

No

It is a condition of publication that authors make their supporting data, code and materials available - either as supplementary material or hosted in an external repository. Please rate, if applicable, the supporting data on the following criteria.

Is it accessible?

Yes

Is it clear?

Yes

Is it adequate?

Yes

Do you have any ethical concerns with this paper?

No

Comments to the Author

Thank you for your response to my comments (reviewer 2) in the previous round of review. I think the manuscript is now much more clearly written, and that the use of functional diversity as a metric for pyrodiversity is now more clearly positioned relative to earlier work (in particular Hempson et al. 2018). I remain somewhat unconvinced about whether the new approach offers a marked improvement on the Hempson et al. approach, and elaborate on this below. In general, however, the clarity of the manuscript has been much improved, and I think that the methods and results are sound.

I was surprised to see that the Hempson et al. approach was indicated as considering only the most recent fire in a pixel in both table s2 and the response to reviewers – this is not the case. Rather, the authors suggest a minimum requirement of data on four or more fire events within in a pixel for a meaningful pyrodiversity estimate to be calculated. Please correct this.

From a quick look at the data provided with this submission, it looks as though nearly half of the watersheds experienced one or zero fires during the study period – what does a pyrodiversity estimate mean if no fires have burned during the period of data collection? The vast areas of zero in Fig. 2 are presumably these areas with no fire data; I'm not convinced that these areas should be included. Furthermore, including these areas in the later analyses leaves me rather unsure what to make of the pyrodiversity peak in response to burn activity in Fig. 4A – while you suggest that there might be an expectation that pyrodiversity would increase monotonically with burn activity (in the response to reviewers), the analyses from this manuscript and those in Hempson et al. suggest the opposite – drier flammable areas (with fewer fires) tend to have higher pyrodiversity than wetter flammable areas (with more fires) – is the intermediate peak a consequence of including areas with no fires and hence no pyrodiversity?

While the manuscript now provides a better comparison between your approach and that of Hempson et al., it still does not critique the differences in the derivation of the respective metrics. As noted above, I'm not convinced that the new approach outperforms the Hempson et al. approach, due to the following reasons. As mentioned, Hempson et al. do not only consider the most recent fire, they 1) quantify traits for all fires during the study period (i.e. the same as the Steel et al. method – but all traits are quantified at the individual fire-level, rather than having some traits measured at the within fire-level [burn severity and patch size] – but considering a patch or the whole fire as 'the fire' means that the same data are used for each method), and then 2) based on a pixel of researcher-determined size (i.e. same as in Steel et al. method), a metric of pyrodiversity is calculated. In Hempson et al., for each pixel, the volume of the four-dimensional convex hull is calculated as the measure for pyrodiversity in that pixel (and is corrected for the positive effect of number of fires on pyrodiversity, through a bootstrapping process using cells from the entire dataset). By contrast, in the Steel et al. approach, pyrodiversity is not calculated in the individual pixel, it is instead calculated at watershed-scale [in this example], which represents a 'community' [watershed] of 'species' [pixels], for which the 'mean multidimensional distance of unique species from the centroid of a community, weighted by abundance' is calculated. If the description of the respective approaches is accepted, then the following contrasts emerge: 1) the Hempson et al. approach will always be able to be applied at a finer-grain than the Steel et al. approach [given adequate data per pixel], because pixels do not need to be grouped into communities [consider revising lines 91-93 and 380-383, which imply that Steel et al. is more

appropriate for fine-scale analyses], 2) the Hempson et al. approach assesses pyrodiversity in each pixel directly from the variability of fire characteristics/traits in that focal pixel – whereas the Steel et al. approach relies on variation among pixels in the community to quantify pyrodiversity [within pixel information is aggregated i.e. mean or weighted mean], and 3) the Steel et al. method requires [at least in the example provided] that trait values are rounded off so that ‘species’ can emerge, whereas the Hempson et al. approach does not require this. To me, these seem to be points where the Hempson et al. approach might outperform the newly proposed approach – but using functional diversity to quantify pyrodiversity in this manner is of course entirely valid if care is taken in how species and communities are delimited [i.e. carefully justify the rounding to delimit species, make sure that communities can be compared in a fair manner – e.g. similar size, adequate data].

The supplementary analyses (thank you for all your effort there) broadly equate functional richness to the Hempson et al. approach, but I think what you’ve done is to use aggregated fire trait values at the pixel-level to then calculate functional richness at the watershed-level – which is quite different to calculating within-pixel functional richness (plus sample size corrections etc.). Accordingly, lines 383-385 do not pertain to the Hempson et al. approach (as implied) – if there was variation in pixel-level burn severity values within the landscape, the Hempson et al. method would capture it because it is calculated at pixel-level, not landscape-level.

Thank you again for all your hard work in responding to my previous comments. The goal of producing a general metric for pyrodiversity is certainly worthwhile, and I think that your ideas would be a valuable contribution to the field.

Decision letter (RSPB-2020-3202.R0)

19-Feb-2021

Dear Dr Steel:

Your manuscript has now been peer reviewed and the reviews have been assessed. The reviewers’ comments (not including confidential comments to the Editor) are included at the end of this email for your reference. As you will see, the reviewers both view the revised manuscript as much improved, but they nevertheless still have some suggestions for revision and improvement, especially Reviewer 2. We would like to invite you to revise your manuscript to address these suggestions.

When submitting your revision please upload a file under "Response to Referees" in the "File Upload" section. This should document, point by point, how you have responded to the reviewers’ and Editors’ comments, and the adjustments you have made to the manuscript. We require a copy of the manuscript with revisions made since the previous version marked as ‘tracked changes’ to be included in the ‘response to referees’ document.

Research ethics:

Use of animals and field studies:

It is a condition of publication that you make available the data and research materials supporting the results in the article (<https://royalsociety.org/journals/authors/author-guidelines/#data>). Datasets should be deposited in an appropriate publicly available repository and details of the associated accession number, link or DOI to the datasets must be included in the Data Accessibility section of the article (<https://royalsociety.org/journals/ethics-policies/data-sharing-mining/>). Reference(s) to datasets should also be included in the reference list of the article with DOIs (where available).

If you wish to submit your data to Dryad (<http://datadryad.org/>) and have not already done so you can submit your data via this link [http://datadryad.org/submit?journalID=RSPB&manu=\(Document not available\)](http://datadryad.org/submit?journalID=RSPB&manu=(Document%20not%20available)), which will take you to your unique entry in the Dryad repository.

Please submit a copy of your revised paper within three weeks. If we do not hear from you within this time your manuscript will be rejected. If you are unable to meet this deadline please let us know as soon as possible, as we may be able to grant a short extension.

Best wishes,
Dr Maurine Neiman
mailto:proceedingsb@royalsociety.org

Reviewer(s)' Comments to Author:

Referee: 1

Comments to the Author(s).

General comments

I provided reviewer comments on a previous draft of this manuscript. The authors have adequately addressed comments from the editor and reviewers, including adding a thorough comparison of multiple methods of calculating pyrodiversity in the main text and supplementary material. The authors have also provided an understandable rationale for not including an analysis of the relationship between pyrodiversity and biodiversity within this manuscript. This paper is well written, introduces a compelling and reproducible method for calculating pyrodiversity, and sets the stage for additional exploration of the relationship between pyrodiversity and ecological patterns and processes.

Specific comments

Line 67. Typo, excess “;”

Line 375. Typo, “evaluates”

Line 381. Typo, “complementary”

Referee: 2

Comments to the Author(s).

Thank you for your response to my comments (reviewer 2) in the previous round of review. I think the manuscript is now much more clearly written, and that the use of functional diversity as a metric for pyrodiversity is now more clearly positioned relative to earlier work (in particular Hempson et al. 2018). I remain somewhat unconvinced about whether the new approach offers a marked improvement on the Hempson et al. approach, and elaborate on this below. In general, however, the clarity of the manuscript has been much improved, and I think that the methods and results are sound.

I was surprised to see that the Hempson et al. approach was indicated as considering only the most recent fire in a pixel in both table s2 and the response to reviewers – this is not the case. Rather, the authors suggest a minimum requirement of data on four or more fire events within in a pixel for a meaningful pyrodiversity estimate to be calculated. Please correct this.

From a quick look at the data provided with this submission, it looks as though nearly half of the watersheds experienced one or zero fires during the study period – what does a pyrodiversity estimate mean if no fires have burned during the period of data collection? The vast areas of zero in Fig. 2 are presumably these areas with no fire data; I'm not convinced that these areas should be included. Furthermore, including these areas in the later analyses leaves me rather unsure what to make of the pyrodiversity peak in response to burn activity in Fig. 4A – while you suggest that there might be an expectation that pyrodiversity would increase monotonically with

burn activity (in the response to reviewers), the analyses from this manuscript and those in Hempson et al. suggest the opposite – drier flammable areas (with fewer fires) tend to have higher pyrodiversity than wetter flammable areas (with more fires) – is the intermediate peak a consequence of including areas with no fires and hence no pyrodiversity?

While the manuscript now provides a better comparison between your approach and that of Hempson et al., it still does not critique the differences in the derivation of the respective metrics. As noted above, I'm not convinced that the new approach outperforms the Hempson et al. approach, due to the following reasons. As mentioned, Hempson et al. do not only consider the most recent fire, they 1) quantify traits for all fires during the study period (i.e. the same as the Steel et al. method – but all traits are quantified at the individual fire-level, rather than having some traits measured at the within fire-level [burn severity and patch size] – but considering a patch or the whole fire as 'the fire' means that the same data are used for each method), and then 2) based on a pixel of researcher-determined size (i.e. same as in Steel et al. method), a metric of pyrodiversity is calculated. In Hempson et al., for each pixel, the volume of the four-dimensional convex hull is calculated as the measure for pyrodiversity in that pixel (and is corrected for the positive effect of number of fires on pyrodiversity, through a bootstrapping process using cells from the entire dataset). By contrast, in the Steel et al. approach, pyrodiversity is not calculated in the individual pixel, it is instead calculated at watershed-scale [in this example], which represents a 'community' [watershed] of 'species' [pixels], for which the 'mean multidimensional distance of unique species from the centroid of a community, weighted by abundance' is calculated. If the description of the respective approaches is accepted, then the following contrasts emerge: 1) the Hempson et al. approach will always be able to be applied at a finer-grain than the Steel et al. approach [given adequate data per pixel], because pixels do not need to be grouped into communities [consider revising lines 91-93 and 380-383, which imply that Steel et al. is more appropriate for fine-scale analyses], 2) the Hempson et al. approach assesses pyrodiversity in each pixel directly from the variability of fire characteristics/traits in that focal pixel – whereas the Steel et al. approach relies on variation among pixels in the community to quantify pyrodiversity [within pixel information is aggregated i.e. mean or weighted mean], and 3) the Steel et al. method requires [at least in the example provided] that trait values are rounded off so that 'species' can emerge, whereas the Hempson et al. approach does not require this. To me, these seem to be points where the Hempson et al. approach might outperform the newly proposed approach – but using functional diversity to quantify pyrodiversity in this manner is of course entirely valid if care is taken in how species and communities are delimited [i.e. carefully justify the rounding to delimit species, make sure that communities can be compared in a fair manner – e.g. similar size, adequate data].

The supplementary analyses (thank you for all your effort there) broadly equate functional richness to the Hempson et al. approach, but I think what you've done is to use aggregated fire trait values at the pixel-level to then calculate functional richness at the watershed-level – which is quite different to calculating within-pixel functional richness (plus sample size corrections etc.). Accordingly, lines 383-385 do not pertain to the Hempson et al. approach (as implied) – if there was variation in pixel-level burn severity values within the landscape, the Hempson et al. method would capture it because it is calculated at pixel-level, not landscape-level.

Thank you again for all your hard work in responding to my previous comments. The goal of producing a general metric for pyrodiversity is certainly worthwhile, and I think that your ideas would be a valuable contribution to the field.

Author's Response to Decision Letter for (RSPB-2020-3202.R0)

See Appendix B.

RSPB-2020-3202.R1 (Revision)

Review form: Reviewer 2

Recommendation

Accept as is

Scientific importance: Is the manuscript an original and important contribution to its field?

Acceptable

General interest: Is the paper of sufficient general interest?

Good

Quality of the paper: Is the overall quality of the paper suitable?

Acceptable

Is the length of the paper justified?

Yes

Should the paper be seen by a specialist statistical reviewer?

No

Do you have any concerns about statistical analyses in this paper? If so, please specify them explicitly in your report.

No

It is a condition of publication that authors make their supporting data, code and materials available - either as supplementary material or hosted in an external repository. Please rate, if applicable, the supporting data on the following criteria.

Is it accessible?

Yes

Is it clear?

Yes

Is it adequate?

Yes

Do you have any ethical concerns with this paper?

No

Comments to the Author

Thank you for taking the time and effort to address my previous comments (reviewer 2) - I think your responses are reasonable and the minor revisions to the manuscript are appropriate. Thanks for the engagement through this review process, and good luck with the next phases of your research, where I anticipate you may get begin to explore relationships with biodiversity.

Line 380: quantify -> quantifying

Decision letter (RSPB-2020-3202.R1)

17-Mar-2021

Dear Dr Steel

I am pleased to inform you that your Review manuscript RSPB-2020-3202.R1 entitled "Quantifying pyrodiversity and its drivers" has been accepted for publication in Proceedings B.

The referee(s) do not recommend any further changes beyond the suggestion in line 380. Therefore, please proof-read your manuscript carefully, make this additional change (and any other necessary minor corrections) and upload your final files for publication. Because the schedule for publication is very tight, it is a condition of publication that you submit the revised version of your manuscript within 7 days. If you do not think you will be able to meet this date please let me know immediately.

To upload your manuscript, log into <http://mc.manuscriptcentral.com/prsb> and enter your Author Centre, where you will find your manuscript title listed under "Manuscripts with Decisions." Under "Actions," click on "Create a Revision." Your manuscript number has been appended to denote a revision.

You will be unable to make your revisions on the originally submitted version of the manuscript. Instead, upload a new version through your Author Centre.

- 1) A text file of the manuscript (doc, txt, rtf or tex), including the references, tables (including captions) and figure captions. Please remove any tracked changes from the text before submission. PDF files are not an accepted format for the "Main Document".
- 2) A separate electronic file of each figure (tiff, EPS or print-quality PDF preferred). The format should be produced directly from original creation package, or original software format. Please note that PowerPoint files are not accepted.

- 3) Electronic supplementary material: this should be contained in a separate file from the main text and the file name should contain the author's name and journal name, e.g. `authorname_procb_ESM_figures.pdf`

All supplementary materials accompanying an accepted article will be treated as in their final form. They will be published alongside the paper on the journal website and posted on the online figshare repository. Files on figshare will be made available approximately one week before the accompanying article so that the supplementary material can be attributed a unique DOI. Please see: <https://royalsociety.org/journals/authors/author-guidelines/>

- 4) Data-Sharing and data citation

It is a condition of publication that data supporting your paper are made available (<https://royalsociety.org/journals/authors/author-guidelines/#data>). Data should be made available either in the electronic supplementary material or through an appropriate repository. Details of how to access data should be included in your paper. Please see <https://royalsociety.org/journals/ethics-policies/data-sharing-mining/> for more details.

If you wish to submit your data to Dryad (<http://datadryad.org/>) and have not already done so you can submit your data via this link <http://datadryad.org/submit?journalID=RSPB&manu=RSPB-2020-3202.R1> which will take you to your unique entry in the Dryad repository.

Once again, thank you for submitting your manuscript to Proceedings B and I look forward to receiving your final version. If you have any questions at all, please do not hesitate to get in touch.

Sincerely,
Dr Maurine Neiman
<mailto:proceedingsb@royalsociety.org>

Reviewer(s)' Comments to Author:

Referee: 2

Comments to the Author(s)

Thank you for taking the time and effort to address my previous comments (reviewer 2) - I think your responses are reasonable and the minor revisions to the manuscript are appropriate. Thanks for the engagement through this review process, and good luck with the next phases of your research, where I anticipate you may get begin to explore relationships with biodiversity.

Line 380: quantify -> quantifying

Author's Response to Decision Letter for (RSPB-2020-3202.R1)

See Appendix C.

Decision letter (RSPB-2020-3202.R2)

18-Mar-2021

Dear Dr Steel

I am pleased to inform you that your manuscript entitled "Quantifying pyrodiversity and its drivers" has been accepted for publication in Proceedings B.

Data Accessibility section

Open Access

Paper charges

Sincerely,

Proceedings B

Appendix A

Response to referees

Comments by the editor and reviewers are repeated followed by our response in *italicized blue text*.

Associate Editor

Board Member: 1

Comments to Author:

The authors of this manuscript proposed a new index measuring pyrodiversity based on fire regime traits. This index is based on the functional diversity definition and is flexible and reproducible in quantifying pyrodiversity. Using this new index, the authors demonstrated the geographical patterns in pyrodiversity in western US, and then explored the influences of fire regimes, climate, topographic roughness, wilderness and human population density on pyrodiversity. The authors also discussed the possible relationships between biodiversity and pyrodiversity. Two reviewers reviewed this manuscript. Although both of them recognized the merits of this manuscript, they also provided many critical comments on the analyses, interpretation of the results, and the presentation of the index. I think these comments are very useful for the authors to improve the current manuscript. I would like to reinforce a couple of points raised by the reviewers.

First, as pointed out by both reviewers, the authors did not put their study in more clear context. Both reviewers suggested that the authors should more clearly compare their new index with previous ones. Reviewer #2 also pointed that the current index is relatively similar to the index proposed by Hempson et al. 2018 in *Ecography*, and used by Beale et al. 2018 in *Ecology Letters*. I agree with both reviewers. I think the authors should carefully compare their index with previous ones in terms of the advantages and disadvantages of different indices, and their outputs in ecological research. Moreover, I think the authors should also try to make the description about the fire regime traits clearer.

Second, the authors could evaluate the relationship between pyrodiversity and biodiversity using both the new index and the previous ones (e.g. the one proposed by Hempson et al. 2018), and compare the results based on different indices.

Third, I think the authors should better present their study in the context of previous studies in the same field. Moreover, writing of the manuscript would benefit from a more careful language proofing.

We are grateful for the thoughtful and detailed comments made by the editor and two reviewers. The suggestions in this review have certainly improved our manuscript. Generally, we have worked to better place this work in the context of relevant literature and more completely compare and contrast our work with previous methods of quantifying pyrodiversity. In particular, we more fully detail how this work builds on Hempson et al. and others, and under which conditions our method provides similar utility vs. where it may be a preferable approach. We have also worked to make descriptions of fire regime traits clearer and have extensively proofed the text for clarity throughout. After much consideration we have decided not to attempt to evaluate the relationship between biodiversity and the various measures of pyrodiversity in this contribution. We believe this would be difficult to do comprehensively as an add-on to this paper but rather deserves the space and focus of a separate analysis. Indeed, we are in the process of collecting multi-taxa biodiversity data to build on the current work and the growing

literature surrounding pyrodiversity. Please see our specific response to reviewer comments below. Line numbers reference the revised (track-change free) version of the manuscript.

Reviewer(s)' Comments to Author:

Referee: 1

Comments to the Author(s)

Summary:

In this paper, the authors present a well thought out, flexible, and reproducible method for quantifying pyrodiversity in the western US based on four fire regime traits. The authors further quantify the relationship between direct and indirect (mediated by burn activity) drivers of pyrodiversity, accounting for climate, topography, and human influence. Finally, the authors present a hypothesis about the relationship between pyrodiversity and biodiversity based on a review of previous studies. This paper is well written, the statistical methods are well-described and seem appropriate, and the focus on reproducibility and flexibility makes this a valuable contribution for future studies. The methodological approach and results are well supported by figures and tables. However, while the hypothesized relationship between pyrodiversity and biodiversity is interesting and of broad appeal, it does not clearly follow from the results of this study and instead seems “tacked on”. In addition, this paper would be strengthened by a quantitative comparison of this new method of calculating pyrodiversity with previous methods, which would enable an assessment of how and where it is superior to these previous methods.

We thank the reviewer for their thoughtful assessment. We have revised the text throughout to make our discussion of pyrodiversity x biodiversity theory more seamless. We have also added addition text in the body of the manuscript and analysis in a supplement to better quantify how our method compares with previous approaches and highlighting limitations of our proposed method.

General comments:

(1) The relationship between pyrodiversity and biodiversity is a central focus of the framing of this paper in the introduction (e.g., lines 64-77, 97-98) and of an entire section in the discussion proposing a new hypothesis (lines 416-490). While the hypothesis is interesting and compelling, it is not clear how it follows from the analyses and results presented in this paper. One suggestion is for the authors to conduct an additional analysis asking how their newly calculated pyrodiversity metric is related to some metric of biodiversity. If this analysis is performed using the same biodiversity data and at the same scale as some of the previous studies referenced, this could also enable comparison between different methods of quantifying pyrodiversity (which would also address my second general comment).

We appreciate that the emphasis on biodiversity may seem odd given the analysis does not include biodiversity data. We have done two things to alleviate this apparent dissonance: 1) throughout we have broadened our discussion of the implications of pyrodiversity to both

biodiversity and ecological process, and 2) we have revised the discussion section on the biodiversity-pyrodiversity relationship to read more as a framework for future research with which this paper and method can facilitate. While testing the relationship between pyrodiversity and biodiversity using different pyrodiversity measures as suggested would be a valuable exercise we believe it would be difficult to do comprehensively as an add-on to this analysis. For example, such an analysis would require biodiversity data from multiple taxa, collected at both fine scales (e.g., occurrence of birds within 100m of a point count sensu Tingley et al. 2016) and broad scales (e.g., species range data across a region or continent sensu Beale et al. 2018). We see the present work as a steppingstone toward such valuable analyses.

(2) The authors do an excellent job of pointing out limitations of previous methods of calculating pyrodiversity and articulating how this new method should represent an improvement. These assertions would be strengthened by a comparison and assessment of different methods of calculating pyrodiversity. Does this new method indeed improve our ability to quantify and understand pyrodiversity? Is this method of calculating pyrodiversity more consistent with our ecological understanding of differences among watersheds? How could this be assessed?

We have 1) better contextualized our proposed method within the literature and expanded the comparison with that of previous work throughout the text (especially lines 91-98, 114-117 and 372-398); 2) added supplementary material laying out the technical differences among some prominent methods (Table S2) and 3) added quantitative comparisons of how different measure of pyrodiversity are correlated to one another when applied to our dataset, their sensitivity to different watershed (sample) sizes, and sensitivity to the resolution of the underlying data (Supplementary material).

(3) I suggest adding a limitations paragraph to the discussion – some limitations of this approach are discussed (e.g., lines 73-75, 405-414), but they are scattered and incomplete. The primary limitation appears to be the duration of MTBS availability, meaning that the “invisible mosaic” is not fully captured. How does this affect estimated pyrodiversity? This seems like it would be particularly problematic for areas that burned very little during this time period or for areas where fire rotations tend to be very long. This also makes it challenging to assess historical pyrodiversity using consistent methods, which seems like it would be particularly important if emulating historical pyrodiversity is the hypothesized management goal for maximizing biodiversity.

Yes, this is an important point. We have added a discussion paragraph that expands on limitation as suggested (lines 372-398). The duration of the fire history data does indeed pose a challenge for evaluating historical pyrodiversity and is a universal limitation of studies that rely on remotely sensed fire patterns. We also more explicitly note that our method measures contemporary rather than historical pyrodiversity (lines 128, 154, and 391).

Specific comments:

Line 58. I found “expanded scrutiny” to be an odd choice of words here. Perhaps “expanded use”?

Replace with “expanded consideration” (lines 57-58)

Lines 73-75, 525-527. It is unclear whether the ability to quantify the “invisible mosaic” is proposed as a strength (due to including decay rate) or a weakness (due to only ~35 years of MTBS data) of this study. Perhaps both?

We see the flexibility to quantify the invisible mosaic as a strength of this approach. The temporal limitations are certainly a challenge (also see response above), but this is true whether the invisible mosaic is explicitly considered or not. We clarified these points in lines 154-167 and 389-398.

Lines 124-202. I suggest describing the traits first and pyrodiversity calculation second.

We’ve reordered this section to improve flow as suggested.

Figure 1. Excellent and helpful figure.

Thanks! We’ve modified Fig. 1c slightly following comments from Reviewer 2.

Lines 163-164. What fire return interval value is assigned to pixels with no fire in the MTBS record?

The difference between the last year of the dataset and the year preceding the start of the dataset. In this case 34 years (2018-1984). Thus, FRI in this case should be considered a minimum estimate as true FRI is greater than this for many areas. This is a necessary limitation of the data, implications of which are now more thoroughly discussed (lines 136-138 and 389-392).

Line 170. Typo, “;29”

Fixed

Line 219. The ability to quantify both direct and indirect effects is very compelling and leads to some interesting results and discussion.

Thanks, we’re happy you found it interesting and clear.

Lines 292-303. I recommend noting that predictors are standardized somewhere in the main text (in addition to in the supplement).

This text has been added to the main text (lines 254-255).

Line 399. Typo “wildness”

Type fixed

Referee: 2

Comments to the Author(s)

This manuscript proposes a new, general approach to quantifying pyrodiversity – and assesses how this index varies in response to environmental and anthropogenic factors. While the use of functional dispersion offers much potential in quantifying pyrodiversity in a generalizable manner, some of the decisions regarding how fire traits are calculated and what sampling units are used are somewhat debatable. I also feel that more could be done to integrate this research into the existing literature. In particular, Hempson et al. 2018 *Ecography* essentially provides the roadmap for this study, with Beale et al. 2018 *Ecology Letters* being the next stop where a general metric of pyrodiversity is used to assess consequences for biodiversity – the apparent premise for this work. That the approach of Hempson et al. and that proposed here could easily each be applied to the respective datasets seems to have been overlooked – it would be interesting to see a more critical assessment of the differences and strengths of each approach.

We thank the reviewer for their detailed comments. The work of Hempson et al. 2018 was certainly a vital pre-cursor to this work and we have revised to highlight where we have built on their study, where their method and ours are complementary, and where our approach differs. This is also true of the broader literature and through our revisions we seek to better contextualize this contribution. We have also added supplementary analysis that directly compares different diversity metrics.

Specific comments

Line 93-95: It is not clear why the authors consider Hempson et al.'s approach to quantifying pyrodiversity as being restricted to coarse-scales – they calculate pyrodiversity for a range of spatial grains from 15 to 120 m grids, and indeed analyse whether pyrodiversity is dependent on spatial grain. Given the availability of fire trait data, there is no limit to the spatial grain at which their approach could be applied. Similarly, it is not clear what is meant by the lack of ability to capture within-fire traits i.e. 'such as variation in burn severity and spatial pattern (i.e. patch size)' – there would seem to be no reason why Hempson et al.'s method could not be applied to the 30 m grain fire trait data in this study. Given that your aim seems to be much the same as that of Hempson et al. (i.e. Hempson et al. abstract: "We present the first generalizable method to quantify pyrodiversity, and use it to address the fundamental questions of what drives pyrodiversity, which fire attributes constrain pyrodiversity under different conditions, and whether pyrodiversity is spatial grain-dependent."), I think you need to do more to justify what a new approach could offer.

Yes, this is a good point. Hempson et al. tested their approach at coarser scales and use different traits than what we use here, but aspects of their method (e.g., using the Convex Hull metric) could potentially be applied at finer scales and using intra-fire traits such as patch size. We've revised lines 91-93 to more precisely state how it's been used rather than implying it is limited to those conditions. Further, we conduct additional comparisons between the metrics of functional

dispersion, functional richness (Minimum Convex Hull), and Simpson's diversity as part of an expanded supplement. Among other incites this new analysis shows that functional dispersion and Simpson's diversity appear to describe variation in burn severity better than functional richness because most watersheds contain the full range of burn severity (also see comments below).

Line 134-135: 'unique combinations of fire regime traits (fire histories) are considered individual species' – please clarify what the sample unit is. Are the units under assessment individual fires, or are they grid cells in a gridded landscape?

The sample unit are grid cells. This has been clarified on line 178.

Lines 133-136: This could be clarified – it goes from species to community and back to species. Am I correct in understanding that: 1) the landscape = a community of pixels (species), and 2) if two pixels share the same fire traits (histories), they are considered to be the same species?

We've attempted to make this clearer in lines 177-181.

What size pixel is used [I see this is clarified later]? How does pixel size – through its effect on the number of weighted-distance-to-centroid values to be averaged (i.e. number of pixels in the community/landscape) – influence the pyrodiversity/FDis value?

We've now clarified this earlier (line 142 and again on line 178), but theoretically FDis could be calculated using other pixel sizes if different datasets were used. More on that below.

From what I can tell, a single position in trait space is calculated for each pixel (species) – as such, there is a single value for each fire trait for each pixel – if so, this means that any variability in that fire trait within the pixel gets summarised to a single value. Accordingly, your pyrodiversity index is very likely to be dependent on pixel size, as a consequence of how much fire trait variation gets summarised to a single pixel-value as pixel size changes. Is this the case?

By analogy – it's like gridding a landscape, recording what trees (fires) you see in each pixel, and averaging/summarising their height, canopy diameter, bark thickness and SLA (four fire traits) – and then calculating FDis using the pixels/grid cells in the landscape. FDis will be robust to the number of pixels ('species richness') in the sense that distance to centroid will be a weighted-average, but the position of that pixel in trait space will be dependent on what trees occur in each pixel and hence the distribution of trait values being summarised. Starting from a single pixel representing the whole landscape, I'd expect the FDis value to increase steadily as pixel size gets smaller and spatial variation in tree communities gets better represented, and to then start to stabilise – but I'd expect this stabilisation to occur sooner in a landscape with low tree diversity and spatial turnover, but much later (i.e. smaller pixel size) for a hyper-diverse region with a mosaic of different vegetation types.

Can this approach be generalised to contexts where the base data are at spatial grains considerably larger than the 30 m pixels in this study?

Thank you for this great question. We've tested how FDis (as well as functional richness [convex hull] and Simpson's diversity) might change when using different data resolutions and our dataset. Specifically, we aggregated (averaged) the data for four fire traits to produce datasets with grain sizes between 30 m and approximately 20 ha. This analysis shows FDis and Simpson's diversity are largely insensitive to scale and this does not vary by the base heterogeneity of the landscape. At least conditional on our dataset, functional richness declines with increasing grain size and does so more rapidly in variable landscapes. These comparisons and more are now in the Supplementary material.

Line 136: Presumably no pixels shared exactly the same histories?

If we had perfect knowledge of past fire history this is theoretically true. However, because of limited fire history data, many share the same fire return interval and seasonality values. We were also compelled to round trait values slightly for computational reasons, so we do ultimately have shared histories (e.g., patch size values of 7.9 and 8.1 log Ha are rounded to 8 when calculating FDis). Text was added to clarify this (lines 150-152), and the degree of rounding/precision can be set by the user in the code (<https://github.com/zacksteel/pyrodiversity/blob/master/code/YosemiteDemo.md>).

Line 144-146: This is the approach used by Hempson et al. [28] – note that they account for some of the effect of outliers by using a non-parametric bootstrap to account for the effect of the number of fires on the volume of the minimum convex hull.

I had forgotten that Hempson et al. employed this approach to mitigate the effect of outliers. We've revised this sentence to be less of an absolute statement about functional richness (line 189).

How does your method control for the positive effect of the number of fires on pyrodiversity?

As I understand Hempson et al.'s approach they used bootstrapping to essentially rarefy their sample of fires. That is to say to account for the tendency of a convex hull to become larger the more fires are sampled. In our case the number of samples (pixels) is independent the number of fires that have occurred in a watershed. Further, FDis is not correlated with sample size/watershed area (see new supplementary analysis) and bootstrapping/ rarefaction is unnecessary. We are, however, interested in the functional relationship between fire activity and pyrodiversity and test how the percent of a landscape burned during 1985-2018 is related to pyrodiversity. This metric is positively correlated with the number of fires (0.66) and shows that the positive relationship between fire activity and pyrodiversity is monotonic rather than linear as one might assume (Figure 4a).

Line 146-148: Note that trait weighting is certainly not unique to FDis – Hempson et al. suggest that future work considers the merits of weighting traits differently while using their method (pg. 896), but neither their study nor yours actually implements differential weighting of traits.

True. We now note this can be done using either approach (lines 190-192).

Fig. 1B & C – these are not particularly informative – one cannot see the full trait surfaces, nor the pixel sizes – and the indirect effects of the three factors you choose to illustrate in C are implied by the arrows going via fire activity.

I'm not exactly clear on what is being suggested with regard to Figure 1C. However, we have made all connections solid lines since the direct effects of climate, topography and human influence are also assessed, although these drivers are most informative as indirect effects mediated by fire activity. The purpose of Fig. 1B is to illustrate that a watershed is parsed into the four trait surfaces used in this analysis. We believe it serves this purpose (and reviewer 1 appears to agree) so have chosen not to modify it.

Line 172: Am I correct in understanding that you used the first and final year of the data availability period to calculate fire return interval? As in, if it burned once during the ~35 year period (e.g. year 10), the fire return intervals would be 10 years (10-start) and 25 years (end-10)? Are you recommending this as a general method, including for areas where fires are rare?

Yes, you are understanding this correctly (except we use the year prior to the first year). This approach effectively calculates the minimum FRI given the limited temporal scope of the data. It is an established way of calculating contemporary fire frequency (e.g., Safford & Van de Water 2013), although accurately calculating FRI in infrequently burned areas requires longer term data. For this reason, we do not recommend using the relatively short time period represented by Landsat data for calculating absolute fire return interval. However, this approach is useful when calculating pyrodiversity as areas where fires are rare will appropriately have lower FDis values (e.g., areas that have not burned since 1985 will have a single FRI value). That being said, our ability to differentiate pyrodiversity among landscapes with long fire return intervals is limited (true of all methods relying on remotely sensed data). Where fire perimeter data are available for a longer period (e.g. California has accurate fire records back to approximately 1908 for large fires) this limitation can be alleviated somewhat. We've expanded on this limitation in lines 135-138 and 389-398.

Safford, H. D., and K. M. Van de Water. 2013. Using Fire Return Interval Departure (FRID) analysis to map spatial and temporal changes in fire frequency on National Forest lands in California. Research Paper PSW-RP-266. USDA Forest Service, Pacific Southwest Research Station, Albany, California, USA.

Line 179-181: Does this not confound burn severity with fire size, to some extent?

Under the hood, FDis is calculated using principle coordinate analysis, which accounts for redundancy among traits and avoids such confounds (Laliberté & Legendre 2010; line 215). In landscapes with few fires, one could exclude either patch size or severity and get a similar FDis value, but their correlation falls as the number of fires in a landscape's history increases making the information included in both increasingly important (Figure 3).

Laliberté E, Legendre P. 2010 A distance-based framework for measuring functional diversity from multiple traits. Ecology 91, 299–305. (doi:10.1890/08-2244.1)

Line 186-194: This is a rather surprising assumption to be making when looking to formulate a generalised measure of pyrodiversity – what if ‘biodiversity’ does not respond by being “most sensitive to recent events but previous fires (the “invisible mosaic”) maintain some influence over landscape pattern and process”? In many fire prone systems, occasional long fire intervals may allow for woody plants to escape the fire-trap, or occasional high intensity/late dry season fires may dramatically restructure vegetation and associated biota – the most consequential events are not necessarily the most recent, and different taxa will respond on vastly different timescales. Furthermore, if a general researcher was interested in pyrodiversity for reasons other than effects on biodiversity (e.g. how does topography shape pyrodiversity), why should biodiversity considerations be built into the metric? When looking to quantify the variability in fire properties within a region, why should the most recent fires be accorded more weight?

The method allows for weighting by recency but does not require it. A user could specify a zero decay rate if all events are assumed equally important (sensu Ponisio et al.) or one if only the most recent is of interest (sensu Hempson et al.). We’ve modified this paragraph (lines 154-167) to clarify.

Line 212-213: Please clarify how trait correlations were made for watersheds with zero fires – or else clarify what is meant by “Specifically, correlations were made among traits for all study watersheds with minimum number of fires ranging from zero to fifteen”

Thank you for pointing this out. This should read one to fifteen. This has been fixed in the text referenced as well as in figure 3.

Line 215-250: I’m not quite sure that I follow the reasoning behind including proportion burned area (i.e. ‘fire activity’) in these models, particularly as it is inherent to at least two of the traits used to quantify pyrodiversity (i.e. fire frequency and patch size). Basically, pyrodiversity should be a measure of how variable fires are within a region – having many large fires is less pyrodiverse than a mix of small and large – having many short fire intervals is less pyrodiverse than a mix of short and long – with the former in each case likely to increase the proportion burned area i.e. ‘burn activity’.

Our principal reason for including burn activity as a variable is that we assume the socioecological variables tested primarily affect pyrodiversity indirectly, as mediated by burn activity. A model without burn activity implicitly assumes the effect of climate, for example, on pyrodiversity is independent of the amount of fire a landscape experienced. Such an inherent assumption seems unlikely and our results illustrate this is not the case. Secondly, we found it useful to test whether pyrodiversity increased monotonically with burn activity or whether it maximizes at some intermediate point.

Line 358-362: What do you mean by an isolated fire event? If pyrodiversity is conceptualised as the level of dispersion in multi-trait space (as you seem to quantify it) – then how can a single

fire trait be used to adequately quantify pyrodiversity? Given the goal of developing a general metric of pyrodiversity, when would you suggest using different numbers of traits?

We wouldn't suggest using fewer traits, but when a landscape has little fire history, using a single trait like burn severity would provide similar results as a multi-trait approach. These results illustrate under what circumstances a multi-trait approach is most important and where previous single-trait approaches may have been adequate – even if conceptually unsatisfying. We've slightly modified this sentence for clarity (line 364-368).

Appendix B

Pyrodiversity Reviewer Comments - R2

NOTE: Responses to reviewer comments are detailed below in blue text following “RESPONSE:”. Any line numbers in the response refer to the Track Changes version of the revised manuscript collated with this document.

Reviewer(s)' Comments to Author:

Referee: 1

Comments to the Author(s).

General comments

I provided reviewer comments on a previous draft of this manuscript. The authors have adequately addressed comments from the editor and reviewers, including adding a thorough comparison of multiple methods of calculating pyrodiversity in the main text and supplementary material. The authors have also provided an understandable rationale for not including an analysis of the relationship between pyrodiversity and biodiversity within this manuscript. This paper is well written, introduces a compelling and reproducible method for calculating pyrodiversity, and sets the stage for additional exploration of the relationship between pyrodiversity and ecological patterns and processes.

RESPONSE: We thank the reviewer again for their original thorough review and are happy to hear they find the revised version improved. We have fixed the remaining typos noted below.

Specific comments

Line 67. Typo, excess “,”

Line 375. Typo, “evaluates”

Line 381. Typo, “complementary”

RESPONSE: The first two typos have been fixed and the third is no longer relevant following changes suggested by reviewer 2.

Referee: 2

Comments to the Author(s).

Thank you for your response to my comments (reviewer 2) in the previous round of review. I think the manuscript is now much more clearly written, and that the use of functional diversity as a metric for pyrodiversity is now more clearly positioned relative to earlier work (in particular Hempson et al. 2018). I remain somewhat unconvinced about whether the new approach offers a marked improvement on the Hempson et al. approach, and elaborate on this below. In

general, however, the clarity of the manuscript has been much improved, and I think that the methods and results are sound.

RESPONSE: We thank the reviewer for their original and subsequent thorough evaluations of this contribution. We have addressed the remaining concerns detailed below. While we would have liked to convince the reviewer that our approach to quantifying pyrodiversity builds substantively on that presented in Hempson et al. (and the broader literature) it is understandable there may be disagreement on this point. Indeed, we expect future contributions to this topic may propose further modifications to build on our approach. As stated in the text, Hempson et al. (as well as Martin and Sapsis, Ponisio et al. and others) made great strides in advancing our understanding of pyrodiversity and we believe this work builds on those previous efforts and creates a further bridge to future advances.

I was surprised to see that the Hempson et al. approach was indicated as considering only the most recent fire in a pixel in both table s2 and the response to reviewers – this is not the case. Rather, the authors suggest a minimum requirement of data on four or more fire events within a pixel for a meaningful pyrodiversity estimate to be calculated. Please correct this.

RESPONSE: We apologize for the error. This has been corrected in Table S2.

From a quick look at the data provided with this submission, it looks as though nearly half of the watersheds experienced one or zero fires during the study period – what does a pyrodiversity estimate mean if no fires have burned during the period of data collection?

RESPONSE: If no fires have occurred within an assessment period there is zero variation in fire history and thus *contemporary pyrodiversity* is zero. This is consistent with our treatment of variation in fire regime traits across space (a watershed or around a sample point). If defining pyrodiversity as variation among fires as is done by Hempson et al. it makes sense that pyrodiversity would be undefined when no fires have occurred, but that is not the case here.

The vast areas of zero in Fig. 2 are presumably these areas with no fire data; I'm not convinced that these areas should be included. Furthermore, including these areas in the later analyses leaves me rather unsure what to make of the pyrodiversity peak in response to burn activity in Fig. 4A – while you suggest that there might be an expectation that pyrodiversity would increase monotonically with burn activity (in the response to reviewers), the analyses from this manuscript and those in Hempson et al. suggest the opposite – drier flammable areas (with fewer fires) tend to have higher pyrodiversity than wetter flammable areas (with more fires) – is the intermediate peak a consequence of including areas with no fires and hence no pyrodiversity?

RESPONSE: We believe that including areas with very low pyrodiversity is important for capturing the full range of variation and fully understanding the relationship between socioecological variables accessed and pyrodiversity.

Ideally, we'd have complete knowledge of a watershed's fire history (back hundreds of years) and be better able to resolve differences among seldomly burned watersheds, but we are limited to the remote-sensing era. Addressing this limitation by looking at only frequently burned watersheds (e.g., greater than X number of fires within a spatial unit sensu Hempson et al.) is a reasonable approach. However, in the current application we believe this would potentially bias the relationships we are trying to test. This decision requires the appropriate caveats and nuanced interpretation (e.g., we are only assessing contemporary pyrodiversity, not absolute pyrodiversity). Comments from both reviewers in the first round of reviews helped us refine our language surrounding this issue.

To the reviewer's specific question about how including unburned watersheds affects our estimate of the effect of fire activity, we would not expect this to change estimates in a significant way. However, we ran the model again without these watersheds to check. The estimated effects of the full model and the reduced model are very similar. Please see a comparison of the parameter estimates below and the relevant marginal effects plot from the reduced model:

Model	Percent Burn Effect (90% CI)	% Burn quadratic
Full	2.448 (2.419, 2.477)	-0.782 (-0.798, -0.766)
Without unburned watersheds	2.225 (2.184, 2.268)	-0.677 (-0.699, -0.656)

While the manuscript now provides a better comparison between your approach and that of Hempson et al., it still does not critique the differences in the derivation of the respective metrics. As noted above, I'm not convinced that the new approach outperforms the Hempson et al. approach, due to the following reasons. As mentioned, Hempson et al. do not only consider the most recent fire, they 1) quantify traits for all fires during the study period (i.e. the same as the Steel et al. method – but all traits are quantified at the individual fire-level, rather than having some traits measured at the within fire-level [burn severity and patch size] – but considering a patch or the whole fire as 'the fire' means that the same data are used for each method), and then 2) based on a pixel of researcher-determined size (i.e. same as in Steel et al. method), a metric of pyrodiversity is calculated. In Hempson et al., for each pixel, the volume of the four-dimensional convex hull is calculated as the measure for pyrodiversity in that pixel (and is corrected for the positive effect of number of fires on pyrodiversity, through a bootstrapping process using cells from the entire dataset). By contrast, in the Steel et al. approach, pyrodiversity is not calculated in the individual pixel, it is instead calculated at watershed-scale [in this example], which represents a 'community' [watershed] of 'species' [pixels], for which the 'mean multidimensional distance of unique species from the centroid of a community, weighted by abundance' is calculated. If the description of the respective approaches is accepted, then the following contrasts emerge: 1) the Hempson et al. approach will always be able to be applied at a finer-grain than the Steel et al. approach [given adequate data per pixel], because pixels do not need to be grouped into communities [consider revising lines 91-93 and 380-383, which imply that Steel et al. is more appropriate for fine-scale analyses], 2) the Hempson et al. approach assesses pyrodiversity in each pixel directly from the variability of fire characteristics/traits in that focal pixel – whereas the Steel et al. approach relies on variation among pixels in the community to quantify pyrodiversity [within pixel information is aggregated i.e. mean or weighted mean], and 3) the Steel et al. method requires [at least in the example provided] that trait values are rounded off so that 'species' can emerge, whereas the Hempson et al. approach does not require this. To me, these seem to be points where the Hempson et al. approach might outperform the newly proposed approach – but using functional diversity to quantify pyrodiversity in this manner is of course entirely valid if care is taken in how species and communities are delimited [i.e. carefully justify the rounding to delimit species, make sure that communities can be compared in a fair manner – e.g. similar size, adequate data].

RESPONSE: We thank the reviewer for further clarification of the Hempson et al. method. We have removed the noted text that implies Hempson et al. is less applicable at fine scales (lines 91-93 & 379-385). Applying a functional dispersion approach is more computationally intensive than calculating functional richness/convex hull and may require rounding when applied the very large datasets such as the one used here. The reviewer makes a good point that this should be done with care and tailored to specific applications.

The supplementary analyses (thank you for all your effort there) broadly equate functional richness to the Hempson et al. approach, but I think what you've done is to use aggregated fire trait values at the pixel-level to then calculate functional richness at the watershed-level – which is quite different to calculating within-pixel functional richness (plus sample size

corrections etc.). Accordingly, lines 383-385 do not pertain to the Hempson et al. approach (as implied) – if there was variation in pixel-level burn severity values within the landscape, the Hempson et al. method would capture it because it is calculated at pixel-level, not landscape-level.

RESPONSE: This aspect of the supplementary analysis isolated the differences in choice of pyrodiversity metric. We believe targeting our comparison to this crucial component of the varied pyrodiversity approaches was the most informative way to proceed – and to address the reviewer's original critiques. This of course is not the only difference between Hempson et al. and our method. We've revised this text (lines 384-388) to more precisely contrast functional dispersion and functional richness rather than the broader Hempson et al. method.

Thank you again for all your hard work in responding to my previous comments. The goal of producing a general metric for pyrodiversity is certainly worthwhile, and I think that your ideas would be a valuable contribution to the field.

RESPONSE: And thank you again for your thorough reviews. They have certainly helped improve this manuscript and my understanding along the way.

Appendix C

Response to reviewers – R3

Reviewer(s)' Comments to Author:

Referee: 2

Comments to the Author(s)

Thank you for taking the time and effort to address my previous comments (reviewer 2) - I think your responses are reasonable and the minor revisions to the manuscript are appropriate.

Thanks for the engagement through this review process, and good luck with the next phases of your research, where I anticipate you may get begin to explore relationships with biodiversity.

Line 380: quantify -> quantifying

RESPONSE: Thank you for taking one last look at our submission and especially for your very thorough and helpful previous reviews. The final edit suggested above has been made.